



# R²D²: Accounting for temporal dependences in multivariate bias correction via analogue ranks resampling

Mathieu Vrac and Soulivanh Thao

Laboratoire des Sciences du Climat et de l'Environnement (LSCE-IPSL), CEA/CNRS/UVSQ, Université Paris-Saclay
Centre d'Etudes de Saclay, Orme des Merisiers, 91191 Gif-sur-Yvette, France

**Correspondence:** Mathieu Vrac (mathieu.vrac@lsce.ipsl.fr)

**Abstract.** Over the last few years, multivariate bias correction methods have been developed to adjust spatial and/or inter-variable dependence properties of climate simulations. Most of them do not correct – and sometimes even degrade – the associated temporal features. Here, we propose a multivariate method to adjust the spatial and/or inter-variable properties while also accounting for the temporal dependence, such as autocorrelations. Our method consists in an extension of a previously

developed approach that relies on an analogue-based method applied to the ranks of the time series to be corrected, rather than applied to their "raw" values. Several configurations are tested and compared on daily temperature and precipitation simulations over Europe from one Earth System Model. Those differ by the conditioning information used to compute the analogues, and can include multiple variables at each given time, a univariate variable lagged over several time steps, or both – multiple variables lagged over time steps. Compared to the initial approach, results of the multivariate corrections show that,

while the spatial and inter-variable correlations are still satisfactorily corrected even when increasing the dimension of the conditioning, the temporal autocorrelations are improved with some of the tested configurations of this extension. A major result is also that the choice of the information to condition the analogues is key since it partially drives the capability of the proposed method to reconstruct proper multivariate dependencies.





# 1 Introduction

Climate model simulations are and will remain the main source of numerical projections to understand and anticipate climate
change consequences. Those projections are performed under various greenhouse gazes emission scenarios, prescribed for
instance within the 5th international "Coupled Models Intercomparison Project" (CMIP5, IPCC, 2013) or the on-going CMIP6
(Eyring et al., 2016), and are widely used by the scientific community investigating climate change and the manifold impacts
of the upcoming climate changes. Indeed, climate changes have been anticipated to affect multiple domains : hydrology and

water resources (e.g., Gleick, 1989; Christensen et al., 2004; Piao et al., 2010), agronomy and crops (e.g., Ciais et al., 2005;
Ben-Ari et al., 2018), ecology and biodiversity (e.g., Brown et al., 2011; Bellard et al., 2012), economy (e.g., OCDE, 2015; Tol,
2018) or human migrations (e.g., Defrance et al., 2017) are examples of domains where expected impacts of climate evolution
can be high and therefore quite problematic for society.

To get robust impact estimations, the climate projections have thus to be as precise and informative as possible. However,
even simulations of the current climate often present statistical biases: their mean, variance, or more generally their distribu-
tions, can more or less largely differ from observational reference datasets (see, e.g., Christensen et al., 2008; Teutschbein and
Seibert, 2012; François et al., 2020, among many other studies). This also means that climate projections for future periods
are also expected to have biases, potentially similar. That is why many impact studies, for current or future climate, have to
rely on "adjusted" climate simulations, obtained via bias correction (BC) methods. Over the last decades, many statistical and

data-science BC techniques have been progressively devised for this specific purpose. The objective of such techniques is to
transform (i.e., "correct", or "adjust") the climate model simulations such that, for a calibration time period, the obtained cor-
rections are equivalent to a reference dataset in terms of one or several targeted statistical features (e.g., means, variances, or
distributions). Simple methods can be used in case the target is only the mean (as the so-called "Delta" or "Anomaly" methods,
e.g. , Xu1999) or the variance (e.g., "simple scaling" Eden et al., 2012; Schmidli et al., 2006). Nevertheless, in general, the

most employed methods are based on the "quantile-mapping" approach (e.g. Déqué, 2007; Gudmundsson et al., 2012) and
its many variants (e.g. Kallache et al., 2011; Vrac et al., 2012; Cannon et al., 2015), whose the target is the whole univariate
distribution (i.e., not only the mean and variance but all moments of higher order, as well as any percentile) of a given climate
variable.

However, if many statistical aspects can be adjusted with such methods, all are only univariate, i.e., related to only one
physical variable at a single location. If multiple variables and/or at multiple locations have to be corrected, the independent
applications of several 1d-BC methods will not modify the intrinsic dependence structure of the simulations to be corrected
(Vrac, 2018). Therefore, if the climate model simulations have biases in their inter-variable and/or inter-site dependencies
(e.g., in their correlations), most of the quantile-mapping and univariate BC techniques will not correct these features and will
basically preserve their biases. This has obviously consequences for the impact models requiring multiple climate variables as
input: if the physical relationships (i.e., the statistical dependencies) of those input variables are not realistic enough, the biases
in the multivariate situations can quickly propagate to the simulated impacts themselves, even if the simulations are adjusted
by 1d-BC methods, (e.g. Boé et al., 2007). More generally in climate sciences, the accurate modelling of dependencies is a



key aspect for proper assessments and projections of compound events and their associated risks (e.g. Leonard et al., 2014; Zscheischler et al., 2018; Bevacqua et al., 2019).

Consequently, some multivariate bias correction (MBC) methods have been recently designed to tackle the issues of the biases in multivariate dependencies. The goal is basically the same as for univariate corrections: find a transformation that makes climate model simulations have the same targeted statistical features as a reference in the calibration period. In this case, the target statistical features does not only include univariate features but also multivariate statistical features such as correlations or the empirical copula. The various MBCs developed so far can be categorized in three main families (Vrac,
2018; Robin et al., 2019; François et al., 2020):

- the "marginal/dependence" approaches, correcting separately univariate distributions and dependences before joining them to provide multivariate corrections (e.g. Vrac, 2018; Cannon, 2017);

- the "conditional successive" methods, adjusting one variable at a time but conditionally on the previously corrected variables to ensure proper multidimensional relationships (e.g. Piani and Haerter, 2012; Dekens et al., 2017);

- the "all-in-one" models, which do not separate the multivariate distribution, neither in marginal/dependence, nor in conditional distributions, but directly transform one multidimensional distribution into another multidimensional distribution (e.g. Robin et al., 2019).

A first intercomparison and critical review of MBC methods has been carried out by François et al. (2020). One major finding was that, although most of the MBC techniques (depending on their hypotheses and configurations) are more or less
able to provide adjusted multidimensional properties, none of them explicitly account for temporal dependence properties. This implies that, although multivariate properties can be correctly adjusted (and sometimes, spatial properties as well, depending on the method), the temporal structure of the data generated by MBC methods is different from that of the model data to be corrected but not necessarily closer to that of the reference data. Therefore, there is a need to improve temporal properties resulting from MBC outputs. Of course, this specific need should not be filled at the expense of the other (marginal, inter-
variable or inter-site) properties.

In the present study, we rely on a recently developed MBC method named $R^2D^2$ to propose an extension allowing us to improve the autocorrelation of the multivariate adjusted data. This $R^2D^2$ extension takes advantage of an analogue-based technique to reconstruct the multidimensional dependence conditionally on temporal sequences of ranks.

The rest of this article is organized as follows: Section 2 describes the reference and model data on which the proposed
$R^2D^2$ extension is evaluated. Section 3 provides a short reminder about the initial $R^2D^2$ approach, the detailed description of its new extension, as well as the experimental design set up for evaluation. Then, results are given and analysed in Section 4, under the underlying focus-question "can the suggested method improve the temporal dependencies without degrading the other (marginal, spatial and inter-variable) properties?". Finally, the main findings are summarised and discussed in Section 5.





## 2   Reference and model data

To perform tests and analyses of the proposed correction method, we will rely on daily temperature at 2 meters (T2) and precipitation (PR) from one run of a global climate model to be corrected on one hand, and from an observation-based reference dataset on the other hand.

The latter corresponds to WFDEI data, which is the WATCH Forcing Data (WFD; Weedon et al., 2011) methodology applied to ERA-Interim data, for the period from 1 January 1979 to 31 December 2016 on a $0.5^o$ x $0.5^o$ spatial grid (Weedon et al.,
2014) over the land-only European region [-$10^o$E, $30^o$E]x[$30^o$N, $70^o$N], corresponding to 4167 gridpoints.

The climate model data to be corrected are extracted – for the same region – from simulations performed by the IPSL-CM5A-MR Earth system model (Marti et al., 2010; Dufresne et al., 2013). A historical run is used for 1979-2005. This is concatenated with a run under RCP8.5 scenario for 2006-2016, hence providing a 1979-2016 time period. Those simulations have an initial $1.25^o$ x $2.5^o$ spatial resolution. To allow comparisons and applications of BC methods, they are then regridded to
the WFDEI spatial resolution with a bi-cubic interpolation for temperature, and a conservative interpolation for precipitation.

Note that only one climate model is used for application and evaluation purposes in the present study. Of course, other models will have other biases that must be corrected differently. However, our goal is not to test the proposed approach on many climate models, but rather to establish a proof-of-concept of the $R^2D^2$ extensions on an illustrative simulations run. We hypothesise that the main general findings obtained on this single model will still be valid for other models and simulations.

## 3   Methods and design of experiments

### 3.1   A short reminder about the $R^2D^2$ method

The proposed methodology relies on – or can be seen as an extension of – the "Rank Resampling for Distributions and Dependences" ($R^2D^2$) bias correction method (Vrac, 2018). $R^2D^2$ consists in 2 steps: first, a univariate BC is performed to adjust the marginal distributions; and then, the empirical copula function (i.e., the dependence structure between the variables of
interest, rid of their marginal distribution) is adjusted. Thus, $R^2D^2$ belongs to the "marginal/dependence" family of multivariate bias corrections (see François et al., 2020, for a description of the other families: ""successive conditional" and "all-in-one")

For the first step, any 1d-BC method can be employed. In Vrac (2018) and in the following of the present study, the "Cumulative Distribution Function - transform" (CDF-t) approach (e.g., Vrac et al., 2012) is used to adjust the marginal properties.

For the second step, $R^2D^2$ uses a "conditioning dimension" (called "reference variable" in Vrac, 2018) from the 1d-BC
results. This univariate 1d-BC time series – and more precisely its ranks – serves as a conditioning to find, within the other 1d-BC variables, the values that have the same ranks association as those in the training reference dataset (details and examples on $R^2D^2$ can be found in Vrac, 2018). Hence, this method relies on a univariate conditioning dimension to generate rank associations, in the same way as an analogue technique (initially developed by Lorenz, 1969) relies on its predictors to generate values. By doing so, this MBC approach allows to reproduce observed multivariate (spatial and multi-variable) dependence
structures, while preserving some temporal properties of the initial simulations via the conditioning dimension.





However, if the temporal features of the conditioning dimension (i.e., one physical variable at one given location) is preserved by construction, this is not necessarily the case for the other variables (i.e., different physical variables and/or spatial locations) and even not the case at all for variables having a weak rank correlations with the conditioning dimension. Therefore, taking advantage of the analogues-based philosophy of $R^2D^2$, several extensions are here proposed to improve the temporal properties of the corrections brought by the initial $R^2D^2$.

## 3.2 Accounting for temporal structures via multivariate ranks conditioning

The main idea of the proposed extensions consists in seeing the $R^2D^2$ approach as an analogue-based method. Indeed, in previous sub-section 3.1, it is clear that the resampling of the multivariate ranks is conditional to a single rank value of the conditioning dimension. In analogues techniques used in the climate literature (e.g., Zorita and von Storch, 1999; Yiou, 2014; Jézéquel et al., 2018, among others), the conditioning (i.e., predictor) variable can be multivariate. In our case, since the purpose of $R^2D^2$ is to correct the dependence structure, we want the notion of analogue situations to only account for the dependence structure and not for the marginal distribution. Hence, the distance between two situations is not computed based on the raw values of the conditioning dimensions but based on their ranks. The best analogue is thus defined as the situation (e.g., day) having the association of ranks the closest to that of the conditioning dimension in terms of Euclidean distance. Here, an extension of $R^2D^2$ is proposed and allows different configurations, all relying on $R^2D^2$ applied conditionally on a multidimensional conditioning dimension:

- $R^2D^2$ conditional on a multivariate information at a given time $t$: The conditioning dimensions in $R^2D^2$ can be chosen freely. They can belong to the set of variables to be corrected, provided as exogenous variables or be a combination of both. There is no restriction neither on the spatial scales of the conditioning dimensions. For instance, as a bivariate conditioning dimension, one could combine a daily NAO index, to provide large-scale information, with the temperature at one gridpoint as a source of small-scale information. Other choices could be the temperature at two given locations, or the temperature and the precipitation at one location, etc.

- $R^2D^2$ conditional on a rank sequence at times $(t-n, t-n+1, \ldots, t)$ of a univariate conditioning dimension: The idea here is about the same as in the previous suggestion but instead of conditioning the ranks resampling on a multivariate conditioning dimension at time $t$, it is on a univariate one (e.g., temperature at a given location, or NAO index) but at several (e.g., $n$) lagged time steps $(t-n, t-n+1, \ldots, t)$.

- $R^2D^2$ conditional on a ranks sequence of a multivariate conditioning dimension: This is a logical combination of the two previous configurations to condition $R^2D^2$ on an information characterising a temporal sequence of multiple variables.

Whatever the configuration, the choice of the conditioning dimension is however not trivial, as it conditions the temporal properties of the model that will be conserved after correction. In the case of a configuration accounting for the rank sequence, the length of the sequence to search the analogues has to be chosen. This length will be referred to as "Block-A" (for "Block-analogue") hereafter. Moreover, in order to avoid discontinuities in the reconstructed final sequence of ranks (and therefore





in the final corrected time series), only a sub-sequence (i.e., shorter than Block-A) of the best analogue sequence is kept. The length of this sub-sequence has also to be chosen and is referred to "Block-K" (for "Block-kept") hereafter. Preliminary tests

(not shown) indicate that Block-A=9 and Block-K=7 are reasonable choices and that the results are only weakly influenced by a slight change of those values.

In the following, 20 different $R^2D^2$ configurations are applied and compared to the reference WFDEI dataset, the plain simulations and the univariate BC results obtained from CDF-t. Those 20 configurations and their notation are given in table 1. For the versions including 5 gridpoints in the conditioning, the locations are chosen to characterise 5 cities: Paris, Madrid,

Stockholm, Rome and Warsaw, spread out over the region. In the same manner – but more automatically – the N gridpoints (N = 100 or 400) in the other versions are chosen to cover uniformly the region of interest.

Note that the configuration using a conditioning with only one physical variable at a single location without accounting for lags (i.e., R.1.1.0) exactly corresponds to the initial $R^2D^2$ method.

Moreover, in practice, the $R^2D^2$ configurations with 400 gridpoints or with 4 167 (i.e., all) gridpoints for the conditioning

dimension provided results equivalent to those from the same configurations but with only 100 gridpoints (not shown). This emphasizes a preliminary result: taking a large number of spatial information is not necessarily needed once a sufficient information is provided. Hence, in the following, the experiments R.400.1.0, R.400.2.0, R.400.1.1 and R.400.2.1 will not be presented, neither the experiments R.4167.1.0, R.4167.2.0, R.4167.1.1 and R.4167.2.1, as they provide results similar to R.100.1.0, R.100.2.0, R.100.1.1 and R.100.2.1, respectively.

### 3.3 Experimental design of the correction schemes

The different configurations of the $R^2D^2$ extensions, as well as the CDF-t univariate BC (referred to as BC1D in the following), are applied and evaluated according to the following 2-fold cross-validation approach: First, the methods are calibrated over the 1979-1997 period and applied to correct the 1998-2016 climate projections for evaluation. Then, they are also applied the other way around, i.e., calibrated on 1998-2016 and applied for evaluation on 1979-1997. Finally, the two 19-year evaluation

periods are gathered to dispose of the whole 38-year time period for evaluation.

Every method is applied on daily values but on a monthly basis, i.e., for each month separately that are joined afterwards. However, evaluations are performed on a seasonal basis – i.e., for each season (DJF, MAM, JJA, SON) separately – to reduce the number of figures and to group similar behaviours.

## 4 Results

In this section, we examine the effects of $R^2D^2$ on the temporal, spatial, inter-variable and marginal properties of the dataset to be corrected. In the rest of the paper, most results are presented for Winter only, but analyses for the other seasons are given as supplementary materials when meaningful. Figure "X" of the supplementary materials will be referred to as Figure SM"X" in the following.





### 4.1 Temporal correlations: are they improved?

Here, we first look at the ability of $R^2D^2$ to reproduce the short-term temporal dependencies of the conditioning dimensions, through the order 1 autocorrelation $\rho$, corresponding to the coefficient of a first-order Auto-Regressive model (AR1).

#### 4.1.1 Temperature temporal correlation

For Winter temperature (Figure 1), the reference dataset shows high AR1 coefficients for the whole region of interest ($\rho(AR1) >$ 0.7). The IPSL dataset and the BC1D dataset also exhibit this characteristic, indicating that the initial model simulations are consistent with the reference. The root mean square error (RMSE) between the AR1 coefficients of the reference dataset, the IPSL dataset or the BC1D dataset is around 0.04. Slight differences can be observed, for instance in Italy and Spain where the $\rho$ values are slightly lower than from the reference dataset.

For R.1.1.0 (conditioning dimension is temperature in Paris, panel 1(d)), the AR1 coefficient of the conditioning dimension from the univariate correction is close to that from the reference data. After the R.1.1.0 correction, the sites whose the temperature autocorrelations are similar to those in the reference are located around Paris. The farther the points are from Paris, the less the $R^2D^2$ correction is able to reproduce the AR1 coefficients observed in the reference. This is explained by two factors. First, the conditioning dimension in the reference and in BC1D are similar in terms of AR1 coefficients. Second, in the reference dataset, there is a strong correlation between the conditioning dimension (the temperature in Paris) and the temperature at sites that are geographically close. Indeed, in $R^2D^2$, at each time step we recopy the rank association observed in the reference dataset, given the rank of the conditioning variable in the BC1D dataset. Hence, for a site close to Paris, the multivariate correction will alter the temperature rank sequence of the reference dataset to make it consistent with the rank sequence of the conditioning dimension in the BC1D dataset. In this case, since temperature in Paris in the BC1D dataset possesses temporal properties similar to the references and because of a strong spatial dependence around Paris, the temporal properties of temperature at a site close to Paris will be, after correction, consistent with the temporal properties of the temperature in Paris in the BC1D dataset and, thus, by transitivity, consistent with the temporal properties of the temperature in Paris in the reference dataset. In the following, we will refer to this phenomenon as the "transitivity effect". Note that variables that are independent of (or only weakly correlated to) the conditioning dimension in the reference dataset have their ranks altered as well but not necessarily in a meaningful way. Indeed, for independent or weakly correlated variables, the rearrangement of the rank sequence is equivalent to a random permutation. Hence, to maximise the transitivity effect, it is needed to select conditioning variables (i) that have similar temporal properties in the reference and in the simulations to be corrected and (ii) that, in the reference dataset, show strong dependencies with the other variables (i.e., site x climate variable) that we want to correct. Based on figure 1(d), it is clear that it is not the case for the temperature in Paris, as already found by Vrac (2018).

However, when increasing the number of sites (R.5.1.0 and R.100.1.0, resp. Figures 1(e-f)) in the conditioning dimension, an amplification of the transitivity effect is visible: the areas where the AR1 coefficients are well reproduced have expanded and are located close to the conditioning sites. Indeed, the mean daily temperature is a relatively smooth signal over this large region and the AR1 coefficients are well represented by the simulations.





Adding precipitation in the conditioning dimension (R.1.2.0, R.5.2.0 and R.100.2.0, respectively figures 1(g) to 1(i)), degrades the AR1 properties compared to having only temperature as conditioning dimensions. It may come from the fact that temperature and precipitation may not be strongly dependent and that conditioning on precipitation to find the value of temperatures for points in the neighbourhood of conditioning sites introduces more noise than signal.

When using lags in the conditioning dimensions, all configurations with lags give similar results in terms of RMSE computed on the AR1 coefficient (RMSE = 0.11) and perform generally better than the configurations without lags. This could be expected since, in this case, short sequences of ranks in the reference dataset are resampled in the $R^2D^2$ corrected dataset. Hence, it mechanically improves the agreement between the reference dataset and the $R^2D^2$ corrected dataset in terms of short-term temporal dependence. This mechanism is the essence of the $R^2D^2$ philosophy, where we recopy, in the multivariate corrected dataset, the rank association that is given by the conditioning dimensions. In the following, we will refer to this mechanism as the "copula effect".

Moreover, the configurations using more sites (R.100.1.1 and R.100.2.1) give slightly better results. The spatial variations of the AR1 coefficients are qualitatively better respected, with lower values of autocorrelation in Spain, UK and Libya compared to the rest of the map. Quantitatively, however, there is a negative bias of about -0.1 on average in terms of AR1 coefficients compared to the reference dataset.

In the end, as the initial temperature simulations have AR1 coefficients similar to those from the references, the IPSL and BC1D simulations show the best temporal properties (Best $R^2D^2$ RMSE = 0.1, BC1D RMSE = 0.04). In terms of temporal correlation, R.1.1.0 (i.e., initial $R^2D^2$ method) and R.2.1.0 give the worst results with only sensible values of the AR1 coefficient around the Paris area. However, the use of a multivariate conditioning dimension and overall the use of a rank sequence into the conditioning dimensions strongly improve the capability of $R^2D^2$ to account for temporal dependence features of the temperature variable. Indeed, the best $R^2D^2$ results are clearly obtained for configurations accounting for lags.

### 4.1.2 Precipitation temporal correlation

For Winter precipitation (Figure 2), the reference dataset exhibits AR1 coefficients with spatial structures smaller than those for temperature. Globally, the model roughly reproduces the spatial distribution of the AR1 coefficients (IPSL RMSE = 0.09) but clearly lacks spatial resolution. The BC1D results exhibit finer spatial structures, for instance in the northern coastline of Scandinavia. However, the BC1D AR1 coefficients are not as good as hose from the IPSL dataset (BC1D RMSE = 0.12). For both IPSL and BC1D, the AR1 coefficients are higher than those for the references in Spain, on the coasts of North Africa and on the northern coasts of Scandinavia. The agreement between the reference data and the raw simulations in terms of AR1 coefficients is not as good for precipitation as for temperature.

When applying R.1.1.0 – the configuration of $R^2D^2$ with the temperature in Paris as univariate conditioning dimension without lag – the AR1 coefficient is not correctly reconstructed. In most areas, the AR1 coefficient is close to zero except in Belgium, Netherlands and North Western Germany where the AR1 coefficient is positive but still negatively biased. This probably reveals a rather weak correlation between the temperature in Paris and the precipitation in the surrounding area. With R.5.1.0, which adds Madrid, Stockholm, Rome and Warsaw as conditioning sites, the precipitation autocorrelation is better





reconstructed around the added conditioning sites. The effect is notably stronger around Warsaw and Stockholm, where the correlation between temperature and precipitation is stronger than in Rome and in Madrid (in general, stronger correlations are observed in Northern Eastern Europe in Winter, not shown). With R.100.1.0, using 100 conditioning sites, the AR1 coefficient reconstruction is improved over all Europe but is still relatively far from the reference.

Adding precipitation in the conditioning dimensions helps improving the precipitation AR1 coefficient since it is likely that the correlation between precipitation in two close sites is stronger than the correlation between temperature in one site and precipitation in the other site. With 100 conditioning sites, geographical features present in the reference dataset start to be visible, for instance, higher AR1 coefficients on the coasts of North Africa and on the northern coasts of Scandinavia. Nevertheless, the order 1 autocorrelations are still biased negatively with respect to the reference dataset. In terms of RMSE,

R.1.100.0, performs slightly better than the BC1D dataset (RMSE(BC1D) = 0.12; RMSE(R.1.100.0) = 0.1) and is on the same level as the raw IPSL simulations (RSME = 0.09), although spatial structures are quite different. The transitivity effect is also limited by the fact that temporal properties of the references and of the BC1D dataset are not so similar. For instance, the AR1 coefficients tend to be lower in the BC1D dataset, both for temperature and precipitation. Such differences necessarily minimise the transitivity effect.

As for the temperature, the configurations of $R^2D^2$ using lags in the conditioning dimensions perform better (RMSE = 0.07), with performances relatively independent on the number of conditioning sites or on the type of climate conditioning dimensions. In this case, those configurations of $R^2D^2$ provide an improvement in terms of RMSE compared to the raw IPSL simulations. Still, the AR1 coefficients are biased negatively compared to the reference dataset: The order 1 autocorrelations are globally not as high as in the WFDEI reanalyses.

Hence, depending on the choice of the conditioning dimensions, $R^2D^2$ can partially recover temporal properties of the reference dataset, especially when conditioning by lagged information via rank sequences. It is however hard for $R^2D^2$ to reconstruct the temporal properties perfectly or even do better than the raw IPSL dataset or the BC1D dataset for temperature, a variable whose the temporality is already well represented in the model simulations. The improvement brought by $R^2D^2$ is more pronounced for precipitation temporal properties: including precipitation itself, or more conditioning sites, or lagged

ranks into the conditioning dimension, provides autocorrelation values and structures more similar to the reference ones than the other datasets do (e.g., raw or BC1D simulations, initial $R^2D^2$ configuration R.1.1.0).

Generally, as seen in this sub-section, although the proposed extensions clearly improve the initial $R^2D^2$ method in terms of temporal correlations, the latter can present some underestimation of the reference temporality, both for temperature and precipitation. This could be linked to an inhomogeneous sampling of the rank associations that are taken from the reference

dataset. This is thus investigated in the next sub-section 4.2.

## 4.2 Reference time sampling & Model chronology agreement

When the conditioning dimension is univariate, continuous, with unique ranks (i.e., no repetitions of values) and belongs to the variables to be corrected, it is the only variable (from the BC1D dataset) that the $R^2D^2$ resampling scheme does not modify. Therefore, in this case, if the number of days is the same in the reference and model dataset, each time step is sampled exactly





once and they are uniformly selected. Hence, in this specific case, $R^2D^2$ reproduces exactly the inter-sites and intervariable empirical copula of the reference, but not the temporal dependencies of the data.

However, when the conditioning has a dimension equal or greater than 2, there is no guarantee that the exact same rank associations exist in the reference dataset. Indeed, the higher the dimension of the conditioning, the less probable it is to find the exact rank association in the reference and in the BC1D dataset. This can come from either (i) a sampling issue: the higher

the dimension, the more points are needed to uniformly sample the space, or (ii) from biases in the dependence structure (biases in the rank associations) of the conditioning dimension in the dataset to be corrected. In this case, $R^2D^2$ uses the rank associations of the reference dataset that is the closest in terms of Euclidean distance. Hence, the rank association of the conditioning dimension in the resulting $R^2D^2$ dataset can be different from that of the BC1D dataset. One consequence is that some time steps (i.e., days in our case) can be resampled several times, while others might not be sampled at all. This can

obviously have consequences on the properties (marginal, inter-sites, spatial, temporal.) of the multivariate corrections.

Therefore, we now analyse the distributions of the time steps that have been selected, since it is an indicator of potential biases introduced by the analogue-resampling scheme in $R^2D^2$.

To reproduce exactly the empirical copula of the reference dataset, each time has to be selected only once. The more uneven the distribution of selected time steps, the more likely it is that the correction has modified the frequency of some situations

with respect to the reference dataset. However, there is not a direct relationship between the unevenness of the distributions and the biases introduced in the correction. For instance, if some rank associations do not appear in the correction, they could have been substituted by a very similar association. In this case the bias introduced would be very small.

The distributions of time steps selected in the reference dataset in January by the different configurations of $R^2D^2$ are shown in Figure 3 (the distributions for April, July and October are provided in Figures SM7-9). As expected, R.1.1.0 presents a

uniform histogram, since it uses a univariate conditioning that permits the sampling of the whole reference time steps. However, this is not the case for the other configurations of $R^2D^2$, which all have dimensions of the conditioning equal or greater than 2 (see Table 1). For the configurations of $R^2D^2$ with only temperature as conditioning dimensions and without time lags (R.5.1.0, R.100.1.0), the sampling is quite uniform. This suggests that the spatial properties of the temperature are quite similar between the reference dataset and the BC1D dataset. When adding the precipitation as conditioning dimensions without time lags

(R.2.1.0, R.2.5.0, R.2.100.0), the histograms are slightly less uniform. This indicates that there can be discrepancies between the references and the BC1D dataset for the spatial dependence of precipitation or the dependence between temperature and precipitation. Finally, When adding time lags in the conditioning dimensions, both for temperature and precipitation, (R.1.1.0, R.1.5.0, R.1.100.0, R.2.1.0, R.2.5.0, R.2.100.0), the distributions of selected times appear also to be less uniform. This is especially true, for R.2.5.0, where we observe a trough in the distribution between days 600 and 700. It indicates that the

modelled rank sequences in this period, rarely appear in the BC1D dataset.

Those elements can help us to interpret the performances of the different configurations of $R^2D^2$ with respect to the reconstruction of temporal, spatial and marginal properties of the temperature and precipitation fields.





Moreover, in order to see how much the different $R^2D^2$ configurations change the temporal structures of the original raw
simulations, for all sites and climate variables, we have computed, the correlation between the ranks of the initial raw simula-
tions and the ranks of the multivariate corrected time series. The closer the correlation is to 1, the less $R^2D^2$ has modified the
temporal structures of the raw simulations. The correlation coefficients for the different sites in Winter are shown in Figures 4
for temperature and 5 for precipitation. The other seasons are shown in Figures SM10-15 of the supplementary materials. By
construction, the CDF-t BC1D mostly conserves the ranks of the raw simulations.

For temperature (Fig. 4), we see that the time series of ranks have been modified substantially by R.1.1.0 (panel (b)). When
the number of geographical sites increases (R.5.1.0 and R.100.1.0, panels (c-d)), we observe the transitive effect and the rank
times series are more correlated to those from raw simulations. It is made possible because the variations of temperature are
spatially smooth and because the references and BC1D data seem to have similar temperature temporal properties.

The transitivity effect is also seen when precipitation is added as a conditioning dimension (R.1.2.0, R.5.2.0, R.100.2.0,
panels (e-g)) or when time lags are added (R.1.1.1, R.5.1.1, R.100.1.1, R.1.2.1, R.5.2.1, R.100.2.1, panels (h-m)). However,
fewer changes are made in the rank time-series when the number of conditioning sites increases. However, for those versions of
$R^2D^2$ with a high number of conditioning sites, the resulting rank time series are slightly more modified (i.e., rank correlation
further away from 1) than with just the temperature as conditioning dimension. It may come from the fact that the higher the
dimension of the conditioning, the more likely the rank sequence of the conditioning dimension has to be modified.


For precipitation (Fig. 5), similar observations can be made. However, the changes in the ranks time series are larger than for
temperature. It can be partially explained by the fact that the transitivity effect is weaker for precipitation. Indeed, precipitation
events occur at local scale and with a spatial correlation radius smaller than for temperature.

Globally, due to the transitivity effect, sites strongly correlated with the conditioning dimension in the reference dataset have
their rank sequences mostly conserved after the correction if the conditioning dimension has similar temporal properties in the
reference and the model. As a consequence, adding more sites in the conditioning dimension generally leads to more regions
that mostly preserve the rank sequences of the model. However, to some extent, this effect can be counteracted by the fact that,
as the dimension of the conditioning grows (e.g., adding rank lags in the conditioning), it becomes harder to find the exact
rank associations in the reference data. It leads to alterations in the rank sequences for the conditioning dimension and for the
sites that are correlated with it, and finally to a potential decrease of the rank correlation between the raw simulations and their
corrections.

### 4.3   Marginal, spatial and intervariable evaluations

As seen previously, some of the proposed $R^2D^2$ extensions allow to adjust temporal dependence structure. However, as the
initial $R^2D^2$ method was designed to bias correct multi-sites and intervariable dependencies in addition to marginal distribution,
one can wonder how the temporal structure improvements – as well as the time sampling features – made by the $R^2D^2$





extensions impact the corrections performed on the other dependencies, i.e., whether or not they degrade the marginal, spatial and intervariable properties. This is the purpose of the sub-sections 4.3.1 (marginal), 4.3.2 (intervariable), 4.3.3 (spatial).

### 4.3.1 Marginal properties

We first check whether the $R^2D^2$ correction schemes are able to reconstruct the marginal properties of the reference dataset through two statistics: the mean and the standard deviation.

For each season and each gridpoint, biases in mean temperatures have been computed and are shown in Figure 6 as boxplots. The associated maps are given in Figures SM16-19. For all seasons, there are clear differences between the reference and the IPSL simulations ($1.58^oC <$ RSME $< 1.84^oC$). The best performances are achieved by BC1D ($0.08^oC <$ RMSE $< 0.2^oC$),

although some light positive or negative biases may appear on some regions, depending on the season (see Figures SM16-19). This strong improvement of CDF-t over the raw simulations was expected as the univariate BC focuses on reconstructing the marginal distribution of the reference.

R.1.1.0 provides similar performances. Since the conditioning dimension is univariate, $R^2D^2$ only performs a permutation of the ranks in time. It then only corresponds to a time reordering of the BC1D correction and does not affect the marginal

distributions.

On average, going from 1 conditioning site to 5, with R.5.1.0, increases the biases in mean ($0.13^oC <$ RMSE $< 0.24^oC$). However, using 100 sites (R.100.1.0) is equivalent to 5 in terms of mean ($0.13^oC <$ RMSE $< 0.22^oC$). Yet, the degradation is more visible when adding precipitation as conditioning and when increasing the number of conditioning sites to 100. For R.100.2.0, the RMSE is between 0.19 and $0.54^oC$ depending on the season, and biases can locally exceed $0.5^oC$, or even $1^oC$ in

Winter over Eastern Europe for instance (Figure SM16, panel(i)). It can be linked to the fact that for R.100.2.0, the distribution of time steps selected is less uniform (Figure 3), hence, modifying the marginal mean values provided by CDF-t.

Similar observations can be made when looking at $R^2D^2$ corrections accounting for lags in the conditioning dimension. Configurations including precipitation have less uniform distributions of selected time steps and have thus higher biases.

The same patterns of biases also occur when looking at the standard deviation of the temperatures (not shown).

For precipitation (Figure 7), the IPSL simulations exhibit important biases for the mean precipitation (0.6 mm/day < RMSE < 1.41 mm/day), with a strong South (negative) to North (positive) gradient of biases (Figures SM20-23, panels (b)). As expected, BC1D greatly reduces the bias in the mean (0.1 mm/d < RMSE < 0.15 mm/d). The effects of $R^2D^2$, on the biases in the precipitation mean is similar to those observed in the biases in the temperature mean. R.1.1.0 configuration provides similar performances as BC1D. With or without time lags, adding precipitation in the conditioning tends to increase the biases. Using

more conditioning sites amplifies the biases as well.

Biases in precipitation standard deviation also follow patterns that are similar to biases in precipitation mean. Biases increase with the numbers of conditioning sites and when precipitation is added.

Generally, for both temperature and precipitation marginal properties, the biases tend to be stronger for $R^2D^2$ configurations that exhibit non-uniform sampling of the time steps selected.





### 4.3.2 Inter-variable correlations

We now evaluate the capability of the different $R^2D^2$ configurations to adjust inter-variable dependencies. We first compute the Pearson correlation between temperature and precipitation for each gridpoint in the corrected dataset. We then compare these correlation values with them from the references. The results are summarised as boxplots of differences in correlation (Figure 8). The associated maps are given for each season in Figures SM24-27 of the supplementary materials. Note that the Spearman

rank correlation analysis provides similar conclusions, although they are perturbed by the very rare rainfall occurrences over North-Africa (not shown), which complicates the analysis of the boxplots. Hence, the Pearson correlation has been preferred.

In the IPSL model and in the BC1D correction, the correlation between temperature and precipitation is weaker than in the reference dataset.

We expect R.1.1.0 to have the best performances with regards to inter-variable rank correlation. Indeed, it has a univariate

conditioning dimension, implying that the empirical copula between temperature and precipitation of the reference data observed during the calibration periods is reproduced almost exactly. In practice, in Figure 8, the boxplots for R.1.1.0 are not exactly 0. It indicates that in the references, the empirical copula between temperature and precipitation is not exactly the same during the two time periods used alternatively for calibration and validation. However, R.1.1.0 (i.e., the initial $R^2D^2$ method) is the main benchmark of the inter-variable evaluation. Indeed, it was designed to adjust the temperature-precipitation depen-

dence of the raw simulations, which is the case since it strongly improves IPSL and BC1D datasets properties. Then, the similar behaviors of the different $R^2D^2$ configurations indicate that their T2 vs. PR correlations are also improved and in a similar way. In other words, for all $R^2D^2$ extensions, including those improving the temporal dependence structures (see sub-section 4.1), the inter-variable correlation is not degraded with respect R.1.1.0 and therefore satisfyingly corrected.

### 4.3.3 Spatial correlations

Finally, we evaluate the spatial correlation by computing the loading values of the first empirical orthogonal function (EOF) obtained from a principal components analysis (PCA) applied on temperature and precipitation separately. For each dataset, we compare the associated loading values with those obtained for the references. The results for Winter and Summer are summarised in Figure 9 as boxplots drawn from differences of loading values between the $R^2D^2$ corrections and the WFDEI references. The associated maps are given as supplementary materials in Figures SM28-31.

For both temperature precipitation, and for all seasons, the raw IPSL simulations have loading values well-centred around those of WFDEI since the median of the differences is close to 0.

Simply by correcting the marginal distribution, BC1D improves the agreement with the reference dataset. Indeed, EOFs are computed from the variance-covariance matrix, which is sensitive to the change in the marginal distributions.

In the $R^2D^2$ configurations, as already explained, the ranks of the non-conditioning variables are shuffled to match those

in the reference dataset during the calibration period. If the inter-site copula is similar during the calibration and validation periods, the $R^2D^2$ configurations should improve the spatial correlations compared to BC1D. This is the case for R.1.1.0, as well as for other configurations, where the median of the difference is close to 0 and where an inter-quartile range of the





differences is narrower than that for BC1D. Interestingly, the configurations with the largest inter-quartile range are those for
which the sampling of the time steps is less uniform (Figure 3), illustrating again the potential impacts of an uneven sampling.

However, many $R^2D^2$ configurations are able to reconstruct spatial properties correctly, at least as well as the initial $R^2D^2$
method (i.e., with a univariate conditioning dimension) that was explicitly designed for it. This is even more visible when
looking at the maps of loading values (Figures SM28-31). Hence, the introduction of additional conditioning information into
$R^2D^2$ – needed to improve temporal properties as seen in section 4.1 – does not degrade much the capability of $R^2D^2$ to adjust
the spatial dependence structures of the climate simulations.

Spatial correlograms are not shown but clearly indicate similar results.

## 5   Conclusions and Discussion

### 5.1   Conclusions

To fill some needs of the climate change impact community, a multivariate bias correction method (MBC) has been pro-
posed in this study. In addition to marginal properties, this MBC is designed to adjust both the inter-site and inter-variable

dependence structures of climate simulations, and at the same time to improve the temporal properties of the corrections. Our
approach is based on the previously existing $R^2D^2$ method (Vrac, 2018) that relied on a univariate "conditioning dimension"
to sample ranks from a reference dataset and, therefore, reconstruct the copula-based spatial and inter-variable dependencies.
The suggested $R^2D^2$ extensions allow resampling ranks given a multivariate conditioning dimension, which could be ranks
of multiple physical variables at a time step $t$, or ranks from a single physical variable but over a sequence of $N$ time steps

$(t-(N-1),\ldots,t)$, or ranks of multiple physical variables over a sequence of $N$ time steps.

Several configurations (i.e., different conditioning dimensions including different sites and climate variables, with or without
lagged information) have been applied to correct daily precipitation and temperature simulations over Europe from a single
climate model run, the IPSL-CM5 Earth System Model (Marti et al., 2010; Dufresne et al., 2013), with respect to the WFDEI
data (Weedon et al., 2014) as references. As the initial $R^2D^2$ approach by Vrac (2018) was able to properly adjust spatial

and inter-variable structure but not the temporal properties of the simulations, the underlying question of the present study
was to understand (i) if the proposed multidimensional conditioning in $R^2D^2$ improves the temporal aspects of the corrections
and (ii) the impact of this conditional resampling on the adjustment quality of the other (i.e., marginal, spatial, inter-variable)
properties. Hence, the various $R^2D^2$ configurations have been evaluated and compared to the raw simulations as well as to
corrections from the univariate BC method CDF-t (e.g., Vrac et al., 2012), first in terms of autocorrelation to characterise

the main temporal aspects, and then in terms of marginal properties, spatial dependences and temperature vs. precipitation
correlations.

For temporal properties, the main conclusions were that including more information (sites and/or lagged ranks) in the con-
ditioning dimension generally improves the reconstruction of the autocorrelation coefficients, both for temperature and pre-
cipitation. However, when the dimension of the conditioning (i.e., the number of variables, sites and lags to condition the

resampling) increases, the distribution of the sampled time steps can be quite different from the uniform one. This has then





consequences mostly on the marginals (i.e., univariate properties), where the mean and standard deviation can have stronger biases for non-uniform sampling. For the other evaluations (spatial and inter-variable properties), although variations in the results are visible depending on the conditioning dimension used, the main conclusion is that the proposed $R^2D^2$ configurations are relatively stable. Thus, in general, the introduction of additional conditioning information into $R^2D^2$ allows improving temporal properties with a good preservation of the capability of the initial $R^2D^2$ to adjust both the spatial and inter-variable dependences of the raw simulations.

### 5.2 Discussion and perspectives

The method suggested in this study is of course upgradeable along different axes.

First, as our goal was not to test the various $R^2D^2$ configurations on several climate models, but rather to establish a proof-of-concept of the $R^2D^2$ extension on an illustrative simulations run, only one climate model has been used for application and evaluation in the present study. Although we hypothesise that the main general findings obtained on this single model application will still be valid for other models and simulations, this will need to be confirmed to generalise and refine our results to more model simulations.

Moreover, the fundamental assumption of $R^2D^2$ is that the spatial and inter-variable copulas (i.e., rank association) is stationary in time, even for future climate projections. This assumption – considering that rank associations act as proxies of physics (Vrac, 2018) and that physics does not change in time – is nevertheless debatable since it needs to be verified in further works. However, it highlights the fact that the conditioning dimension has to be carefully chosen to be relevant, both to drive (condition) the correction of the properties of interest, but also to translate the potential changes that may happen in future climate and that would impact the corrections.

More generally, the choice of the conditioning dimension is a key-element of the $R^2D^2$ method. Indeed, as seen in this study, what is corrected or not by $R^2D^2$ is partially driven by the chosen conditioning information. Thus, testing alternative conditioning dimensions could also be of interest for future work, to bring additional physical/geographical information, valuable to generate proper multivariate corrections. Those alternative conditioning – e.g., including North Atlantic Oscillation (NAO) or other indices, characterisation of the circulation or other covariates – have nevertheless to be determined according to the specific region of interest, the climate variables to be corrected, etc. This adaptation of the conditioning to the application is a requirement to inject the relevant and suited physical information into $R^2D^2$.

Of course, if a "good" conditioning must optimize the $R^2D^2$ correction of some statistical properties, it mainly has to optimize the properties that are the most useful for the users of the corrections. In other words, the choice of the $R^2D^2$ configuration has to be tailored for the end-users of the simulations. It is thus very important for the end-users to know which properties are essential to be corrected in order to design the $R^2D^2$ configuration the most appropriate for their specific application. Indeed, if many statistical features of the simulations are to be corrected, it is not clear that one single configuration will be able to correct all properties. For some regions and sets of climate variables, this can happen, but in other cases it might be needed to prioritize the most essential ones and then choose the associated $R^2D^2$ configuration.



Finally, trying to correct multiple statistical properties at the same time remains a difficult challenge, as adjusting one often
modifies another one. Additionally, one can wonder what is kept from the raw climate simulations if a correction is performed
to adjust many statistical aspects. Hence, when applying a multivariate bias correction method with a configuration allowing
to modify (explicitly or implicitly) several properties, a compromise has always to be searched, in order to balance, on the one
hand, the level of correction needed to make the simulations useful for the application of interest, and, on the other hand, the
climate model signal preserved by the applied correction method. This is the only way to make the (M)BC useful in practice
and physically reliable.

*Code and data availability.* The R$^2$D$^2$ code (Vrac and Thao, 2020), specifically developed for this study and used to adjust the depen-
dence structure of the 1d-bias correction data, is available as an R package "R2D2" under the CeCILL licence, and is downloadable at
https://doi.org/10.5281/zenodo.3786848. This package includes the source code, example data and a user manual. The CDF-t code applied
to perform univariate bias correction has been taken from https://cran.r-project.org/web/packages/CDFt/index.html. The IPSL-CM5A-MR
model data simulations as part of the CMIP5 climate model simulations can be downloaded through the Earth System Grid Federation por-
tals. Instructions to access the data are available here: https://pcmdi.llnl.gov/mips/cmip5/data-access-getting-started.html The WFDEI data
used as reference in this study can be accessed following the instructions described in Weedon et al. (2014) or in the following web-page
http://www.eu-watch.org/data_availability.

*Author contributions.* MV had the initial idea of the method, developed the associated code and made the figures. ST wrapped the R package.
MV and ST performed the analyses and wrote the article.

*Competing interests.* The authors declare that no competing interests are present.

*Acknowledgements.* This work has been supported by the EUPHEME and CoCliServ projects, both part of ERA4CS, an ERA-NET initiated
by JPI Climate and co-funded by the European Union (grant no. 690462). We acknowledge the World Climate Research Programme's
Working Group on Coupled Modelling, which is responsible for CMIP, and we thank the IPSL climate modeling groups for producing and
making available their model output. For CMIP the U.S. Department of Energy's Program for Climate Model Diagnosis and Intercomparison
provides coordinating support and led development of software infrastructure in partnership with the Global Organization for Earth System
Science Portals. Finally, MV would like to thank P. Yiou (LSCE) for fruitful discussions about analogues.





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



| Identification name | conditioning dimensions | lags accounted for | dimension of ref. var. |
|---|---|---|---|
| R.1.1.0 | Temperature in the Paris gridpoint | No (Block-A=1; Block-K=1) | 1 |
| R.1.2.0 | Temp. & Prec. in the Paris gridpoint | No (Block-A=1; Block-K=1) | 2 |
| R.5.1.0 | Temperature in 5 gridpoints | No (Block-A=1; Block-K=1) | 5 |
| R.5.2.0 | Temp. & Prec. in 5 gridpoints | No (Block-A=1; Block-K=1) | 10 |
| R.100.1.0 | Temperature in 100 gridpoints | No (Block-A=1; Block-K=1) | 100 |
| R.100.2.0 | Temp. & Prec. in 100 gridpoints | No (Block-A=1; Block-K=1) | 200 |
| R.400.1.0 | Temperature in 400 gridpoints | No (Block-A=1; Block-K=1) | 400 |
| R.400.2.0 | Temp. & Prec. in 400 gridpoints | No (Block-A=1; Block-K=1) | 800 |
| R.4167.1.0 | Temperature in (all) 4167 gridpoints | No (Block-A=1; Block-K=1) | 4 167 |
| R.4167.2.0 | Temp. & Prec. in(all) 4167 gridpoints | No (Block-A=1; Block-K=1) | 2 x 4 167 = 8 334 |
| R.1.1.1 | Temperature in the Paris gridpoint | Yes (Block-A=9; Block-K=7) | 1 x 9 = 9 |
| R.1.2.1 | Temp. & Prec. in the Paris gridpoint | Yes (Block-A=9; Block-K=7) | 2 x 9 = 18 |
| R.5.1.1 | Temperature in 5 gridpoints | Yes (Block-A=9; Block-K=7) | 5 x 9 = 45 |
| R.5.2.1 | Temp. & Prec. in 5 gridpoints | Yes (Block-A=9; Block-K=7) | 2 x 5 x 9 = 90 |
| R.100.1.1 | Temperature in 100 gridpoints | Yes (Block-A=9; Block-K=7) | 100 x 9 = 000 |
| R.100.2.1 | Temp. & Prec. in 100 gridpoints | Yes (Block-A=9; Block-K=7) | 2 x 100 x 9 = 1 800 |
| R.400.1.1 | Temperature in 400 gridpoints | Yes (Block-A=9; Block-K=7) | 400 x 9 = 3 600 |
| R.400.2.1 | Temp. & Prec. in 400 gridpoints | Yes (Block-A=9; Block-K=7) | 2 x 400 x 9 = 7 200 |
| R.4167.1.1 | Temperature in (all) 4167 gridpoints | Yes (Block-A=9; Block-K=7) | 4 167 x 9 = 37 503 |
| R.4167.2.1 | Temp. & Prec. in (all) 4167 gridpoints | Yes (Block-A=9; Block-K=7) | 2 x 4 167 x 9 = 75 006 |

**Table 1.** Summary of the 20 $R^2D^2$ configurations tested. The identification name is organized in the following way: The first number indicates the number of gridpoints used for the conditioning dimension of $R^2D^2$; the second one corresponds to the number of variables considered at each gridpoint for the conditioning dimension (here, "1" indicates "only temperature", "2" means "temperature and precipitation"); and the third number indicates if some lagged (i.e., temporal) information is used by $R^2D^2$ ("0" means "no lag used", "1" means "lags used"). "Block-A" corresponds to the block size (i.e., lags length) used for the analogues search and "Block-K" to the block size that is kept from the selected analogues of size Block-A.

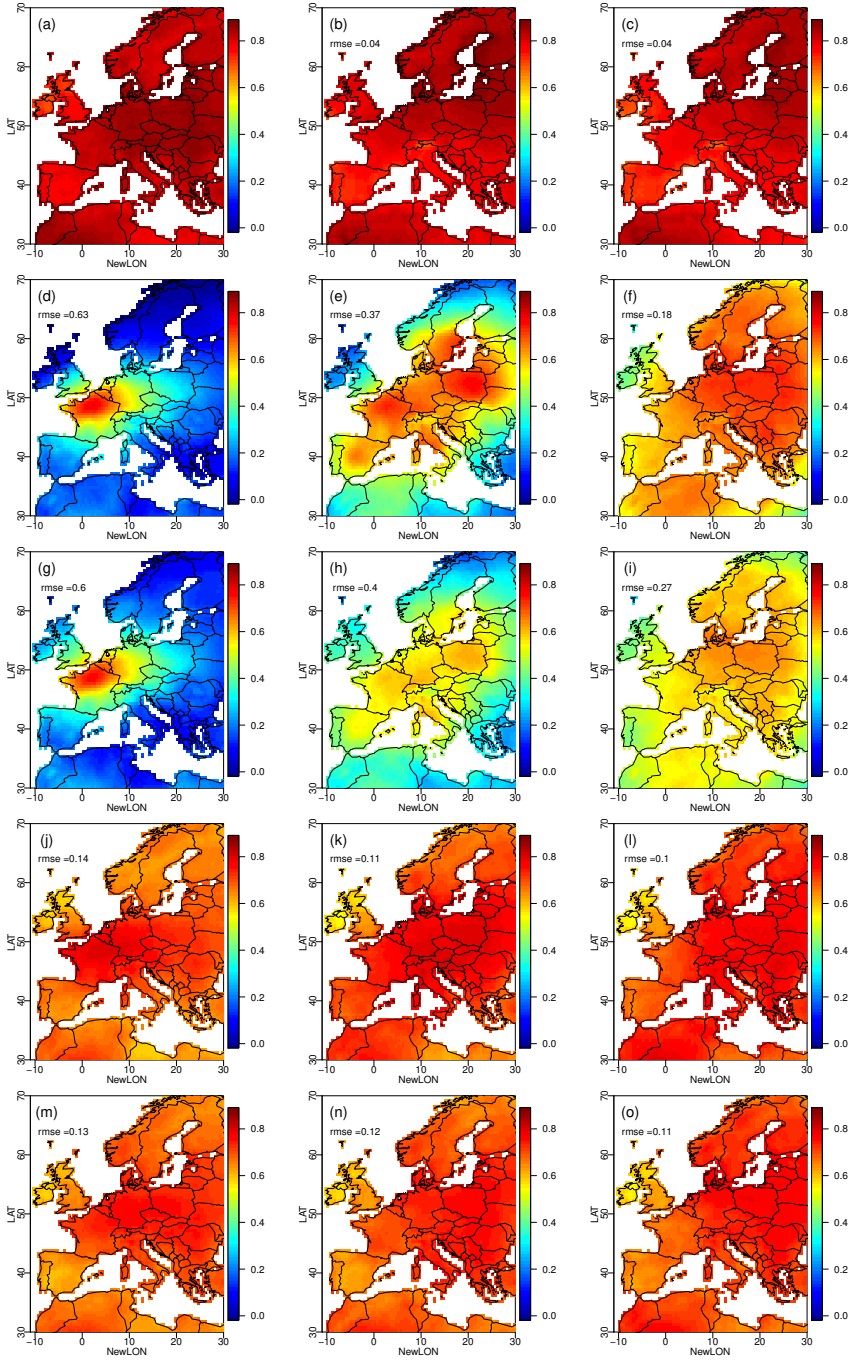

**Figure 1.** Maps of order 1-day temperature autocorrelations for Winter over the 1979-2016 period, for (a) WFDEI, (b) IPSL raw simulations, (c) 1d-bias correction (CDF-t), (d) R.1.1.0, (e) R.5.1.0, (f) R.100.1.0, (g) R.1.2.0, (h) R.5.2.0, (i) R.100.2.0, (j) R.1.1.1, (k) R.5.1.1, (l) R.100.1.1, (m) R.1.2.1, (n) R.5.2.1, (o) R.100.2.1. In other words, 2nd row: results for temperature as conditioning dimension (for different numbers of locations) and without accounting for lags; 3rd row: same but for temperature and precipitation together as conditioning dimension; 4th and 5th rows: same as 2nd and 3rd but with lags accounted for. For (b-o), the RMSE value, computed over the whole domain between WFDEI autocorrelations and those from the model or corrected data, is indicated.



**Figure 2.** Same as Figure 1 but for winter precipitation autocorrelations. Note that, here, precipitation is never used alone as conditioning dimension.



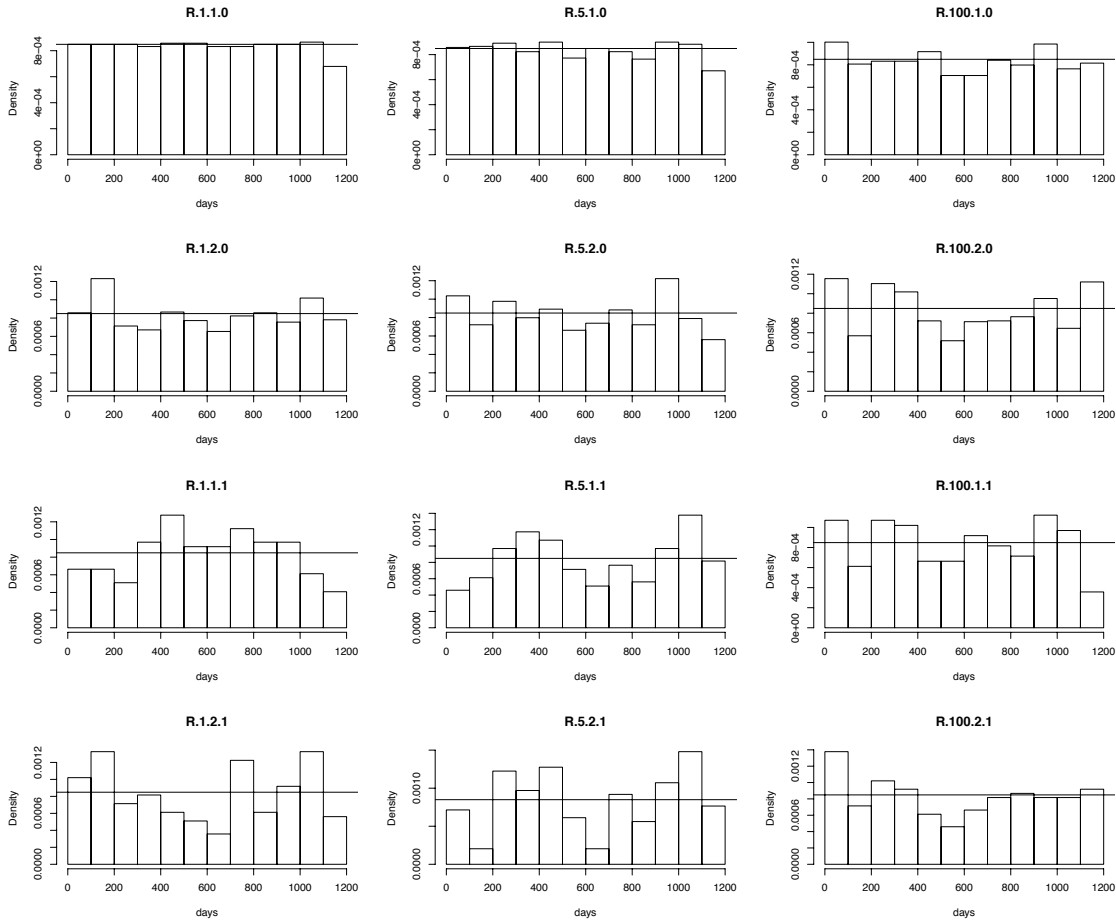

**Figure 3.** Distributions of time steps selected in the reference dataset in January by the different $R^2D^2$ configurations. The equivalent histograms for April, July and October are provided as supplementary materials in Figures SM7-9.

610

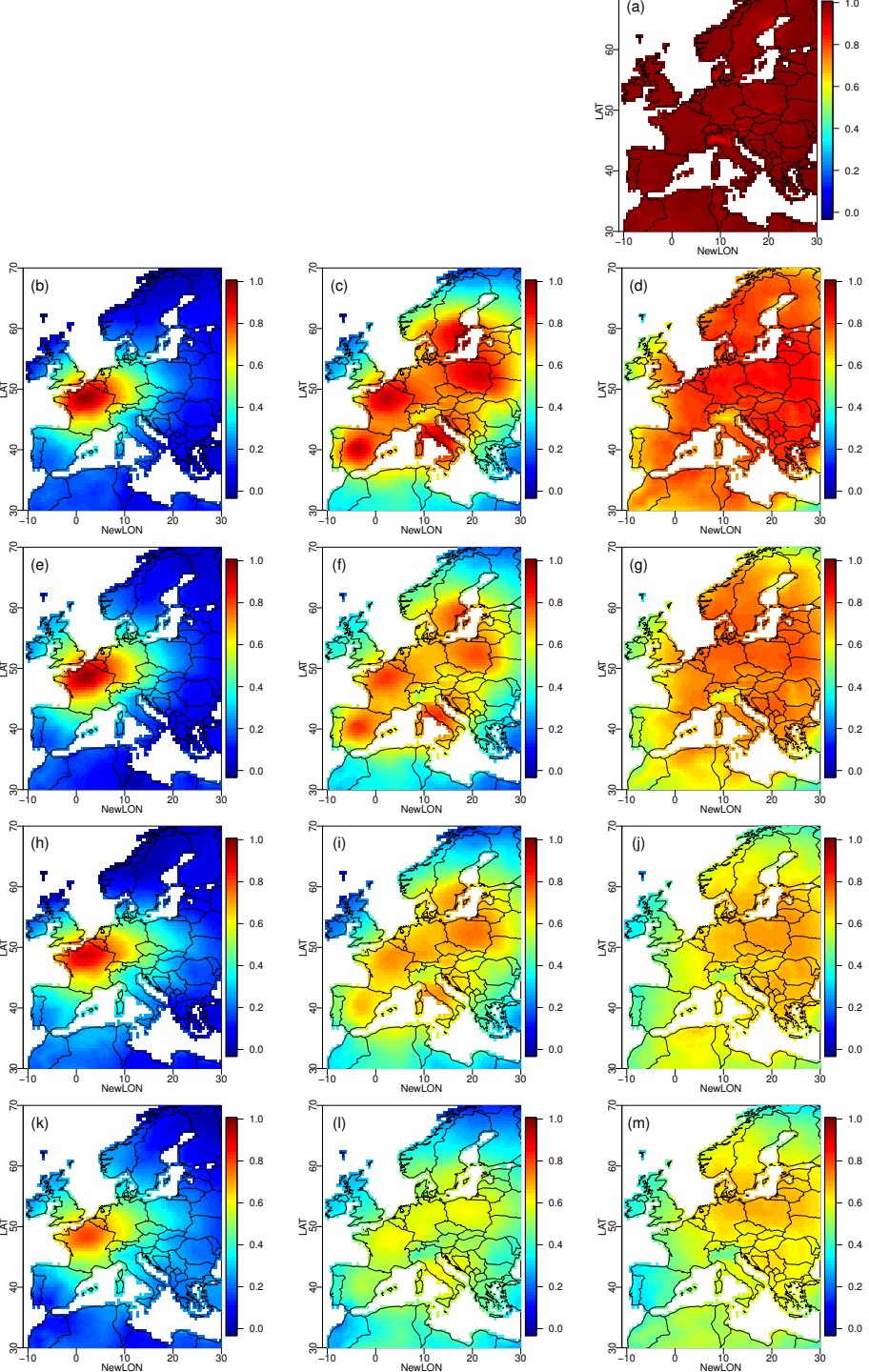

**Figure 4.** Maps of spearman (rank) correlations calculated for each gridpoint in winter over 1979-2016 between the initial climate model temperature simulations and their corrections by (a) 1d-BC, (b) R.1.1.0,(c) R.5.1.0, (d) R.100.1.0, (e) R.1.2.0, (f) R.5.2.0, (g) R.100.2.0, (h) R.1.1.1, (i) R.5.1.1, (j) R.100.1.1, (k) R.1.2.1, (l) R.5.2.1, (m) R.100.2.1. The results for the other season are provided as supplementary figures.



**Figure 5.** Same as Fig. 4 but for precipitation.





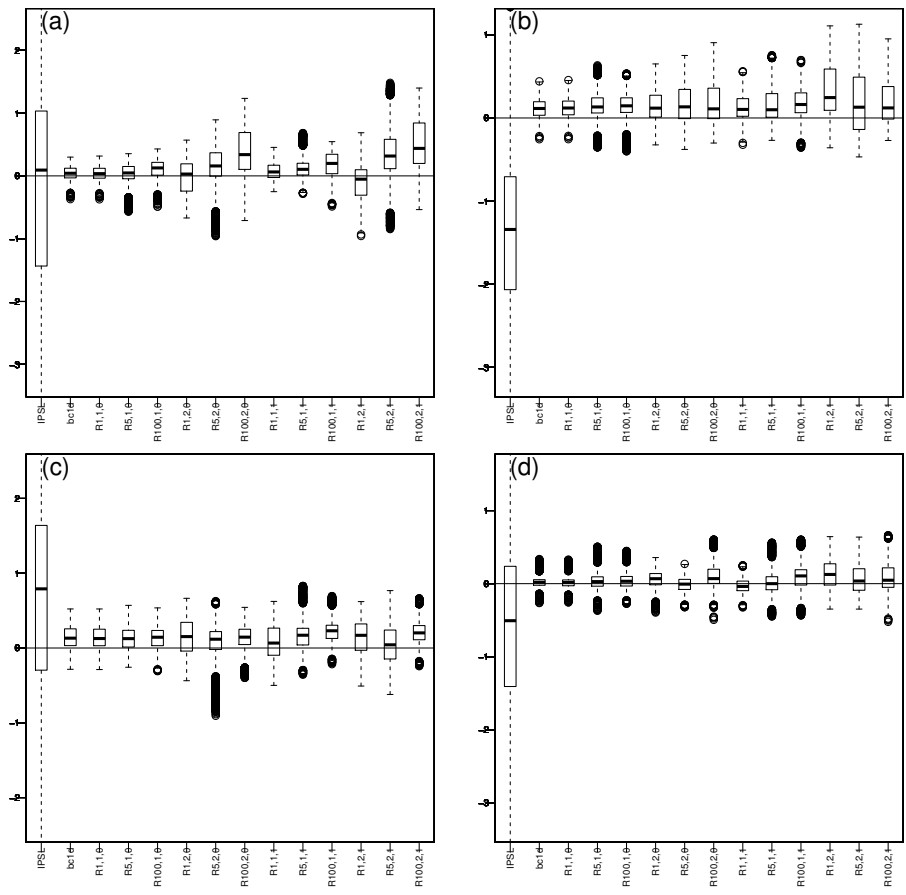

**Figure 6.** Boxplots of differences in mean temperature per gridpoint with respect to WFDEI (i.e., mean(model or BC) minus mean(WFDEI): (a) winter, (b) spring, (c) summer, (d) fall. The associated maps are given in Figures SM16-19.





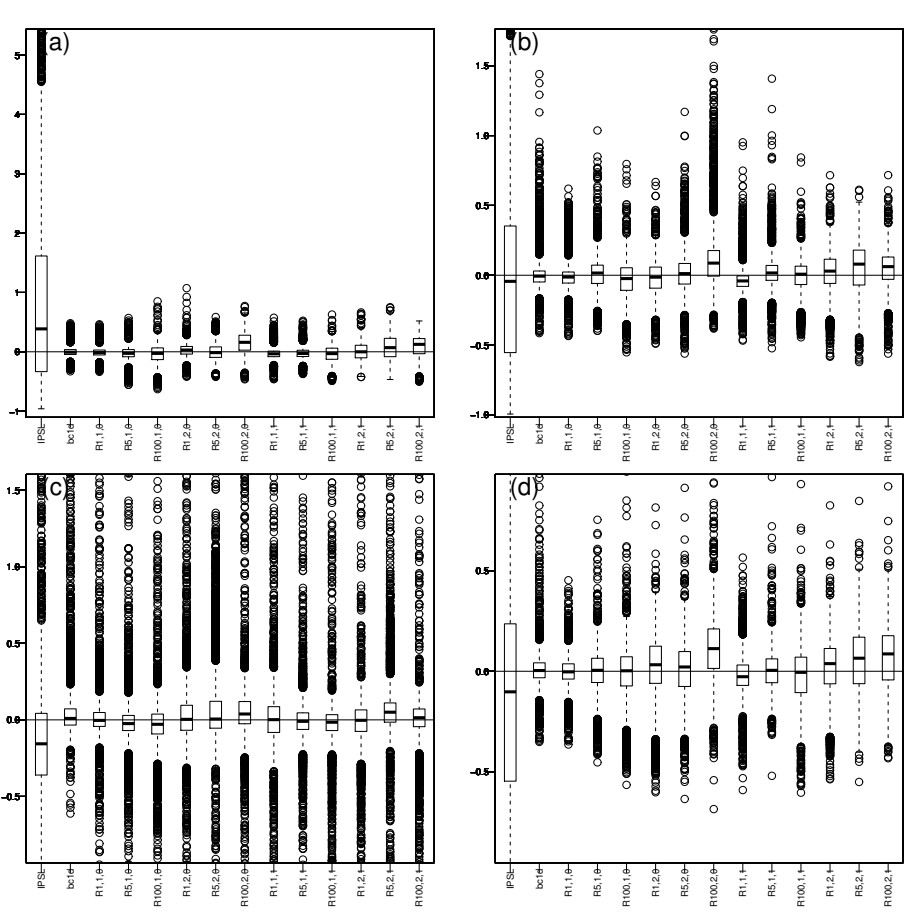

**Figure 7.** Same as Figure 6 but for precipitation. The associated maps are given in Figures SM20-23.



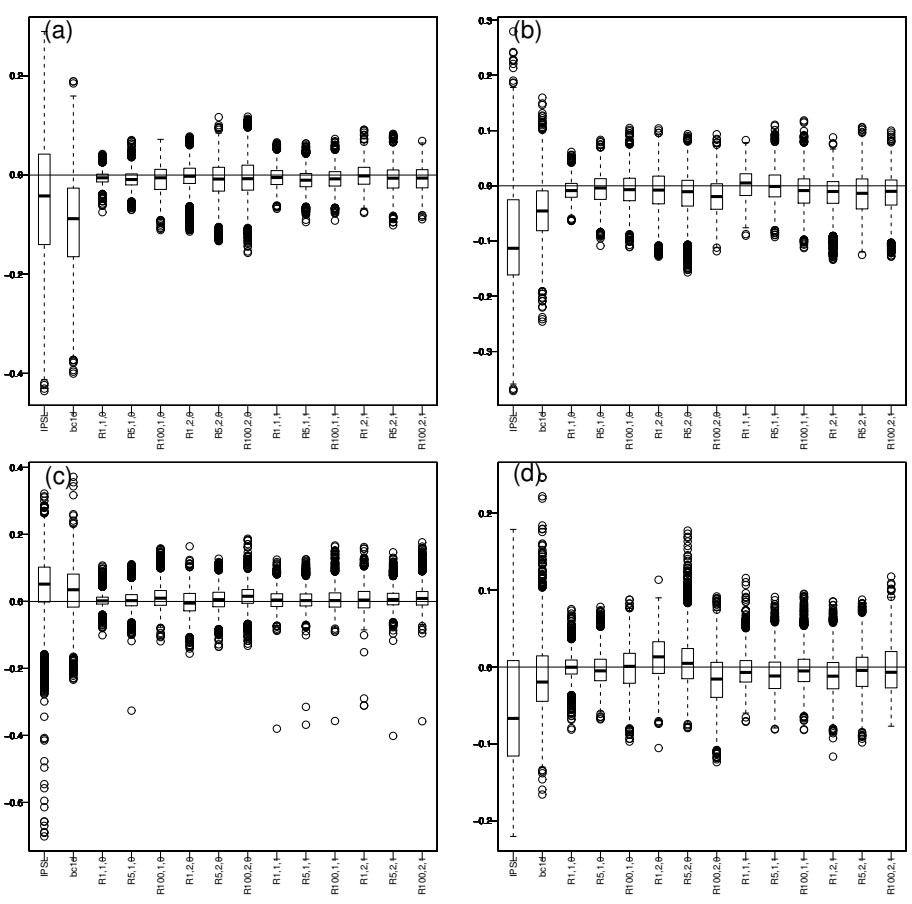

**Figure 8.** Boxplots of differences in Temperature vs. precipitation Pearson correlations between WFDEI and the different datasets (IPSL, 1d-BC IPSL and the $R^2D^2$ configurations) over 1979-2016 in (a) winter, (b) spring, (c) summer and (d) fall. The associated maps are given for each season as supplementary materials.



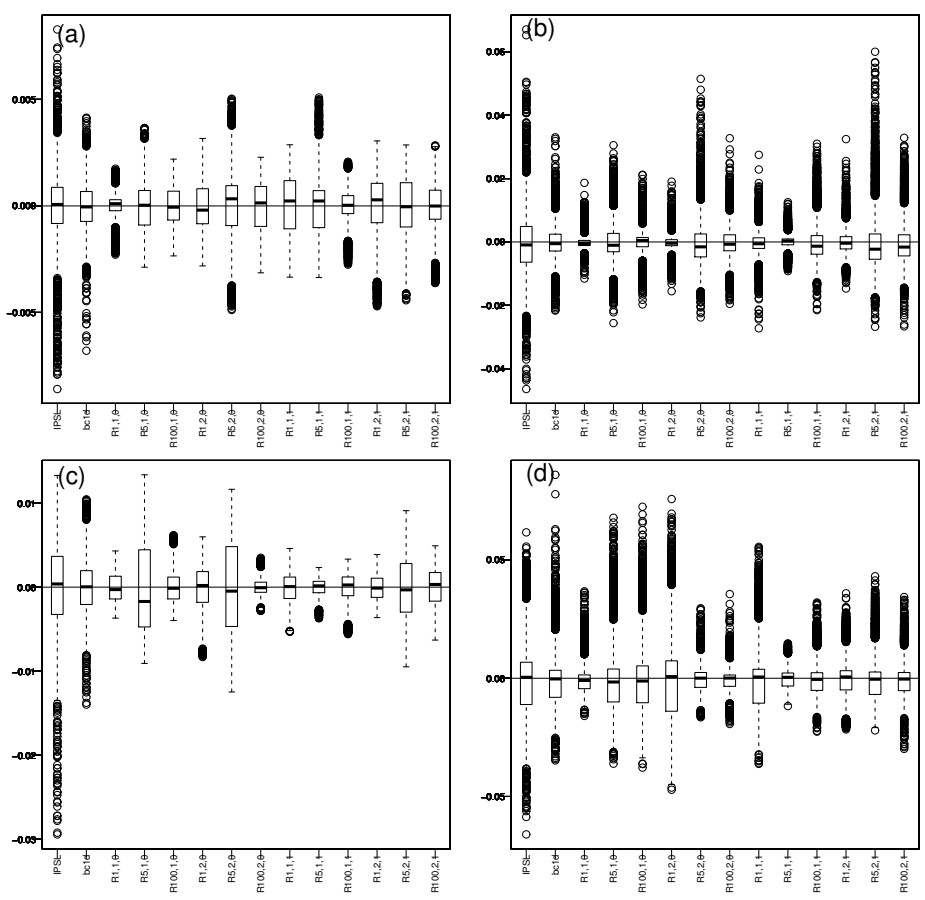

**Figure 9.** Boxplots of differences in loading values for the first EOF (EOF1) between model or corrected data, and WFDEI (i.e., EOF1(model or BC) minus EOF1(WFDEI)). Panels (a) and (c) are for temperature, (b) and (d) for precipitation, for winter (a and b) and summer (c and d). The associated maps are given as supplementary materials.