# Peer review of "$R^2D^2$ v2.0: Accounting for temporal dependences in multivariate bias correction via analogue ranks resampling"

_Geoscientific Model Development, 2020_

## Short Comment (SC1) · 8 Jul 2020

Dear authors,

in my role as Executive editor of GMD, I would like to bring to your attention our Editorial version 1.2:

https://www.geosci-model-dev.net/12/2215/2019/

This highlights some requirements of papers published in GMD, which is also available on the GMD website in the 'Manuscript Types' section:

http://www.geoscientific-model-development.net/submission/manuscript_types.html

In particular, please note that for your paper, the following requirement has not been met in the Discussions paper:

- "The main paper must give the model name and version number (or other unique identifier) in the title."

Please add a version number for $R^2D^2$ in the title upon your revised submission to GMD.

Yours,

Astrid Kerkweg

---

## Referee Comment (RC1) · Verena Bessenbacher (Referee) · 29 Jul 2020

general comment:

The paper presents an interesting advancement in multivariate bias correction, that aims at incorporating the temporal dependence structure on top of the inter-site and inter-variable dependence structure. In general, I think it is important to advance on statistical methods that are able to deal with highly structured climate model output, including different variables, spatial context and temporal autocorrelation. I am not aware of any other method with similar aims, however, my knowledge of the literature in the area of bias correction is not too strong.

The authors use only one run from one global climate model. Since there are no set hyperparameters that tailor this method specifically to IPSL, there is no evidence that this method is not generalisable. As a proof of concept, in my opinion, it is sufficient to work with one climate model only.

comments on the method:

The authors describe from line 437 that adding more sites to the conditioning dimension improves the results, however, at the cost of uneven distribution of the selected timesteps. As stated in line 290, this does not necessarily lead to a larger bias introduced. However, the authors do not recommend choosing a conservative "compromise setting" with not too little and not too many sites used, to hedge both against missing transitivity effect and uneven timestep distribution. I wonder if such a cautionary note should be introduced, that the readers are aware that you can "overfit"? Or do the authors disagree with this statement?

I am a bit confused on how the authors treat the precipitation data. Since precipitation is not normally distributed and often zero, the ranks should exhibit many ties when searching for analogues. How is this problem approached? Additionally, in section 4.3.1. the marginal properties of the corrected data are characterised by the difference of the mean and the standard deviation to the reference dataset (Figure 7). I assume these statistics could be more insightful if the precipitation would be divided into two variables for analysis: (1) if it rained and (2) how much it rained. This could show more in detail possible biases that are corrected or introduced by this method.

Just as a side note, the concatenation of a historical run with an RCP8.5 run (described from line 87) could in worst case lead to a jump in the data. Maybe the authors would like to check whether this is the case and whether this impacts their autocorrelation results.

comments on the results:

The results of the temporal correlation are only shown for winter. I assume this is because the correlations have shown to be higher in winter (see for example line 243), or because the authors argue that this is the most difficult case, but a clear statement arguing for looking at the winter case only in Figure 1 and 2 would be desirable from my point of view. Additionally, since this is a proof-of-concept for a newly developed method, it is interesting for the community to see whether the improvements seen for the temporal autocorrelation are constant or variable throughout the different seasons.

Throughout the text, six results are mentioned that are not supported by the figures, indicated with a "(not shown)" after the result. I would argue the Appendix has space enough for all these results. I would especially be interested in the analysis of the standard deviations of the marginals and the reasoning why the spearman correlation couldn't be used (the paper describes some problems in North Africa, line 381).

comments on the text readability:

The authors manage to describe quite clearly and elegantly their approach throughout the paper, with some minor readability flaws: From line 142 the concept of "Block-A" and "Block-K" is explained. The explanation of the "Block-K" concept and why it is necessary could be a bit more elaborated. In line 189 and further down in the text the word "recopy" is used. What does it mean? How is it different from "copy"?

technical corrections:

line 16 gazes -> gas

line 18 use of "climate change" and "climate changes" in the same sentence. Clarify difference between these terminologies or rewrite

line 142 the sentence starting with "Moreover" is a bit convoluted

line 217 you never use the term "copula effect" again

consider adding links to the respective subfigures in section 4.1.2 for clarity and easy

lookup

line 232 hose -> those

line 268 & section 4.2 do reference temporality and model chronology refer to the same thing? If so, consider only using one of the terms

line 437 particularly clear and concise summary

―――――――――――――――

---

## Referee Comment (RC2) · Sylvie Parey (Referee) · 26 Aug 2020

The paper handles the difficult issue of climate model bias correction extensions to tackle the adjustment of temporal, spatial and inter-variable dependency biases. Based on a previously proposed technique by the authors, named R2D2, different variants are designed and tested with one climate model simulation and one reference dataset for temperature and precipitations. The methodology is meaningfully exposed and the results are clearly commented. This constitutes an important and valuable contribution to this question of bias correction, which remains a key issue in climate impact studies. Especially, the temporal evolution of the variables is often an important feature for

climate impact models, which even require a finer timestep than daily (up to hourly).

My main comments are the following: - When using CDFt for the univariate bias correction of rainfall, how are no-rain days handled? This may explain the degradation in the temporal autocorrelation after 1dBC, which is not seen for temperature. The adjustment of the number of rainy days, besides that of the rainfall amount, is one of the main problems in impact studies. - Section 4.2 discusses the rank association between the corrected and the raw model simulation outputs. Indeed, there is no reason why the model should reproduce the observed chronology, since it represents another sequence of variability. However, some level of correction of the model chronology may be needed (in association with the variables interdependencies for example, as treated here). The problem then is the lack of an adequate reference. . . - In section 4.3.2, it is noted that "the empirical copula between temperature and precipitation is not exactly the same during the two time periods used alternatively for calibration and validation". This may be worth exploring further: how stable is the association with time? How long has the considered period to be in order to faithfully estimate the association? How can climate change alter it? This problem is properly mentioned in the discussion, but could be raised here. - In the conclusion, the improvements in the temporal evolution brought by the variants of R2D2 proposed here are emphasized, but still, they do not reach the rather good level obtained with 1dBC. Hence, in many impact studies, a correct chronology of the variables is of major interest, especially if long duration occurrences of extreme conditions are of concern. This could be stated more clearly.

Minor comments: p2 l36: "whose the target is the whole univariate distribution": I would write "whose target"; the same applies to p7 l184, p9 l263 p4 l101: ""successive conditional": too many quotes p8 l232: "as hose from the IPSL dataset" as those p10 l302: "Finally, When adding time lags in the conditioning dimensions": no capital letter needed for "when" p10 l302-303: "both for temperature and precipitation, (R.1.1.0,R.1.5.0, R.1.100.0, R.2.1.0, R.2.5.0, R.2.100.0): isn't it rather R.1.1.1, R.1.5.1, R.1.100.1, etc. . . ? The same stands for the following sentence: "This is especially true,

for R.2.5.0," R.2.5.1? p13 l378: "We then compare these correlation values with them from the references." With those from the reference? Supplementary Material: Blank lines appear in some maps, what does it mean? The scales in figures 17, 18 and 19 differ, which make it difficult to compare. This is the same for the precipitation figures. It may however be difficult to use the same scale for all plots, but when possible, it's easier for the reader.

---

## Author Comment (AC1) · 14 Sep 2020

**Responses to Executive Editor (Astrid Kerkweg) about "R²D²: Accounting for temporal dependences in multivariate bias correction via analogue ranks resampling"**

Mathieu Vrac and Soulivanh Thao

Laboratoire des Sciences du Climat et de l'Environnement (LSCE-IPSL), CEA/CNRS/UVSQ, Université Paris-Saclay
Centre d'Etudes de Saclay, Orme des Merisiers, 91191 Gif-sur-Yvette, France

Comment:

*Dear authors,*

*in my role as Executive editor of GMD, I would like to bring to your attention our Editorial version 1.2:*

*https://www.geosci-model-dev.net/12/2215/2019/*

5    *This highlights some requirements of papers published in GMD, which is also available on the GMD website in the 'Manuscript Types' section: http://www.geoscientific-model-development.net/submission/manuscript_types.html*

*In particular, please note that for your paper, the following requirement has not been met in the Discussions paper: - "The main paper must give the model name and version number (or other unique identifier) in the title."*

*Please add a version number for R2D2 in the title upon your revised submission to GMD.*

10    *Yours,*

*Astrid Kerkweg*

Answer:

Indeed, this was missing. As the method presented in this submission relies on an extension/improvement of the $R^2D^2$ method

15    by Vrac (2018), it is then its version 2. The title is now:

"$R^2D^2$ *v2.0: Accounting for temporal dependences in multivariate bias correction via analogue ranks resampling*"

**References**

Vrac, M.: Multivariate bias adjustment of high-dimensional climate simulations: the Rank Resampling for Distributions and Dependences (R2D2) bias correction, Hydrology and Earth System Sciences, 22, 3175–3196, https://doi.org/https://doi.org/10.5194/hess-22-3175-2018, 2018.

20

---

## Author Comment (AC2) · 14 Sep 2020

**Responses to reviewer 1 (Verena Bessenbacher) about "R$^2$D$^2$: Accounting for temporal dependences in multivariate bias correction via analogue ranks resampling"**

Mathieu Vrac and Soulivanh Thao

Laboratoire des Sciences du Climat et de l'Environnement (LSCE-IPSL), CEA/CNRS/UVSQ, Université Paris-Saclay
Centre d'Etudes de Saclay, Orme des Merisiers, 91191 Gif-sur-Yvette, France general comment:

*The paper presents an interesting advancement in multivariate bias correction, that aims at incorporating the temporal dependence structure on top of the inter-site and inter-variable dependence structure. In general, I think it is important to advance on statistical methods that are able to deal with highly structured climate model output, including different variables, spatial context and temporal autocorrelation. I am not aware of any other method with similar aims, however, my knowledge of the literature in the area of bias correction is not too strong.*

Answer:

We thank the reviewer for her careful reading and interesting comments. We tried to take them into account as much as possible in the updated manuscript.

Comment:

*The authors use only one run from one global climate model. Since there are no set hyperparameters that tailor this method specifically to IPSL, there is no evidence that this method is not generalisable. As a proof of concept, in my opinion, it is sufficient to work with one climate model only.*

Answer:

Indeed, even though the suggested correction method is specifically applied to the IPSL model simulations in the present article, the methodology is applicable to any climate model.

comments on the method:

*The authors describe from line 437 that adding more sites to the conditioning dimension improves the results, however, at the cost of uneven distribution of the selected timesteps. As stated in line 290, this does not necessarily lead to a larger bias introduced. However, the authors do not recommend choosing a conservative "compromise setting" with not too little and not too many sites used, to hedge both against missing transivity effect and uneven timestep distribution. I wonder if such a cautionary*

*note should be introduced, that the readers are aware that you can "overfit"? Or do the authors disagree with this statement?*

Answer:

This is a wise recommendation. This is now done (on lines 494-496 of the track-changes manuscript and lines 479-481 of the revised one):

"*Hence, in practice, we recommend a "compromise setting" in the choice of the conditioning dimension, with not too few and not too many sites. Such a choice would prevent both from missing transitivity effect and from uneven timestep distributions.*"

Comment:

*I am a bit confused on how the authors treat the precipitation data. Since precipitation is not normally distributed and often zero, the ranks should exhibit many ties when searching for analogues. How is this problem approached?*

Answer:

This is fair question. We acknowledge that our explanations were not necessarily sufficient to understand this point.

Regarding the non normal distribution of precipitation, based on the present methodology, this is not an issue since $R^2D^2$ works explicitly with ranks and ranks associations (i.e., the copula functions). Hence, whatever the distribution (i.e., Gaussian or not) of the continuous part of the data, the ranks provide an independent information.

Regarding the zeros, this is a common and recurrent problem: how to deal with ranks of zero precipitation values? In $R^2D^2$, this question arises when applying $R^2D^2$ with precipitation alone as conditioning dimension, and particularly when there is a single conditioning location. In this case, when the conditioning value is 0, its ranks is 1, and there are many time steps sharing the same first rank. For the moment, the proposed $R^2D^2$ algorithm will select the first time step having this rank of conditioning dimension to perform the rank resampling of the other variables/locations. Hence, the order in which the dataset is searched to find the conditioning rank value will influence the selected time step and may thus potentially also influence the results, especially with precipitation alone at a unique site as conditioning dimension. However, although the same procedure is applied when using multiple conditioning dimensions (i.e., multiple conditioning locations and/or variables) whose at least one contains precipitation, the conclusion only partially holds. Indeed, multiplying the dimensions to condition the rank resampling will reduce the choice of the time step that can be considered for resampling, hence reducing the weight of the ties in the conditioning information.

Other choices could be made to deal with tied ranks: last or median time step (instead of first) could be selected, or even randomly drawn time step (among those with a common distance to the condition information). Nevertheless, this is not investigated in the present paper – since it is a proof of concept – and left for future work.

To clarify those points in the manuscript, the text has been modified (on lines 160-173 of the track-changes manuscript and 155-168 of the revised manuscript) as follows:

"*In case of ties for the choice of best analogues, the proposed $R^2D^2$ algorithm selects the first time step having the minimal Euclidean distance. Hence, the order in which the dataset is searched to find the analogue ranks will influence the selected time*

*step and may thus potentially also influence the results. For example, ties might occur when including daily precipitation as a conditioning variable, particularly when precipitation at a single location is the only conditioning dimension. Indeed, there are potentially many time steps where it did not rain at the conditioning location. All of these time steps would all have the same rank (in this case 1). This means that some rank combinations that are also compatible with the rank of the conditioning dimension (in this example no-rain at the conditioning location) will not be present in the corrected dataset. Hence, some rank associations can be either under-represented or over-represented in the corrected dataset compared to the reference dataset because of ties in the conditioning dimensions. However, when using multiple conditioning dimensions (i.e., multiple conditioning locations and/or variables), the number of candidates for the best analogue with the same exact minimal Euclidean distance decreases, hence reducing the effect of the ties in the time-series of the conditioning dimensions. Other choices could be made to deal with tied ranks: last or median time step (instead of first) could be selected, or even randomly drawn time step (among those with a common minimal distance to the condition information). Nevertheless, this is not investigated in the present paper and left for future work."*

Comment:

*Additionally, in section 4.3.1. the marginal properties of the corrected data are characterised by the difference of the mean and the standard deviation to the reference dataset (Figure 7). I assume these statistics could be more insightful if the precipitation would be divided into two variables for analysis: (1) if it rained and (2) how much it rained. This could show more in detail possible biases that are corrected or introduced by this method.*

Answer:

Thank you. This is an important remark. To follow the reviewer suggestion, the analyses for the marginal of precipitation have been redone but, this time, separating precipitation occurrences and conditional precipitation intensities given rainfall occurrence. Hence, previous figure 7 (corresponding to boxplots of relative differences of mean precipitation with respect to the mean reference precipitation) is now replaced by:

– a figure showing boxplots of the relative differences of rainfall occurrence probability with respect to that of the reference;

– and, in the same figure, the boxplots of the relative differences of the mean conditional rain intensity (given rainfall occurrence) with respect to that of the reference dataset.

Note that, for each new boxplot figure, the associated maps are also given for each season as supplementary materials.

In addition, boxplots of relative differences of the standard deviations computed from conditional precipitation intensities are also given in new figures of the supplementary materials.

To account for these changes, the text now reads (on lines 400-414 of the track-changes manuscript and lines 387-400 of the revised one):

*"For precipitation, univariate biases are investigated in separating occurrences of rainfall and conditional intensities given*

*rainfall occurrences. Hence, Figure 7 displays, for the four seasons, boxplots of the relative differences of the probabilities of rainfall occurrence with respect to that the reference data (7(a-d)), as well as the boxplots of relative differences of the mean conditional intensities given rainfall occurrences, wrt that of the reference data (7(e-h)). The associated maps are given as supplementary materials (Figures SM21-24 for occurrence probabilities and SM25-28 for mean conditional intensities). Rainfall occurrences are defined as precipitation values $> 0.1mm/day$ to get rid of the drizzle effect present in many climate*

*model simulations (e.g., Dai, 2006; Kjellström et al., 2010; Teutschbein and Seibert, 2012). Generally speaking, the effects of $R^2D^2$ on the occurrence and conditional mean precipitation biases are similar to those observed on the mean temperature: (i) the R.1.1.0 configuration provides similar performances as BC1D; (ii) with or without time lags, and with or without adding precipitation in the conditioning, increasing the number of conditioning sites may lead to relatively higher biases, both for occurrence probability and intensity. However, including precipitation itself in the conditioning does not amplify the precipitation*

*biases and can even reduce them depending on the season.*

*Biases in standard deviations for conditional precipitation values are also given for information (Figures SM29) and coincide with results for means.*"

Comment:

*Just as a side note, the concatenation of a historical run with an RCP8.5 run (described from line 87) could in worst case lead to a jump in the data. Maybe the authors would like to check whether this is the case and whether this impacts their autocorrelation results.*

Answer:

To perform our analyses, we selected the same ensemble member (identified with number r1i1p1) both for the historical simulations and the projection. By convention, as recommended by the CMIP5 Data Reference Syntax (DRS) and Controlled Vocabularies document (https://pcmdi.llnl.gov/mips/cmip5/cmip5_data_reference_syntax.pdf?id=32), the identification number of a member stays consistent between different types of experiment. In particular, for RCP simulations, this means that for a given member, the simulations for the projection period have been initialised with the historical simulations of the same member so that the projections are a direct continuation of the historical simulations. This ensures a continuity from historical to RCP simulations. Additionally, the transition in the prescribed external forcing from the historical period to the projection period is quite smooth, which means that no abrupt jump should be present in the data. Nonetheless, we looked at the time series with basic statistics and performed visual investigations of the time series. None of the performed assessments indicated a jump in the data. For the sake of reproducibility, we have added, in the "Reference and model data" section, the identification number of the IPSL-CM5A-MR model run that we used (lines 88-90 of the track-changes manuscript and 86-88 of the revised one):

"*Historical simulations from the ensemble member "r1i1p1" is used for 1979-2005. This is concatenated with simulations under RCP8.5 scenario made from the same ensemble member for 2006-2016, hence providing a 1979-2016 time period.*"

comments on the results:

*The results of the temporal correlation are only shown for winter. I assume this is because the correlations have shown to be higher in winter (see for example line 243), or because the authors argue that this is the most difficult case, but a clear statement arguing for looking at the winter case only in Figure 1 and 2 would be desirable from my point of view. Additionally, since this is a proof-of-concept for a newly developed method, it is interesting for the community to see whether the improvements*

*seen for the temporal autocorrelation are constant or variable throughout the different seasons.*

Answer:

The results for the temporal correlation are only shown for winter **in the manuscript** but results for the other seasons are provided **in the supplementary material (Figures SM1-6)**. Hence, presenting only winter results in the core manuscript allows having a reasonable number of figures, while all complementary results are also accessible to readers in the supplementary information. This is mentioned at the beginning of Section 4: "*In the rest of the paper, most results are presented for Winter only, but analyses for the other seasons are given as supplementary materials when meaningful.*" To emphasise it even more, an additional sentence has been added in the manuscript (lines 286-287 of the track-changes version and 280-281 of the revised one):

"*This is also the case for the other seasons, as shown in supplementary materials in figures SM1-3 for temperature and in figures SM4-6 for precipitation.*"

Comment:

*Throughout the text, six results are mentioned that are not supported by the figures, indicated with a "(not shown)" after the*

*result. I would argue the Appendix has space enough for all these results. I would especially be interested in the analysis of the standard deviations of the marginals and the reasoning why the spearman correlation couldn't be used (the paper describes some problems in North Africa, line 381).*

Answer:

As detailed in previous answers, to follow suggestions given by the reviewer, the supplementary materials document has increased to include more additional figures. Therefore, some "not shown" mentions have been deleted. For example, as indicated earlier, the results concerning the standard deviations (for both temperature, and conditional precipitation) are now given for the 4 seasons in supplementary materials (figures SM20 and SM29 ). The text has been modified to read:
- Line 393-394 of the track-changes version and 386 of the revised one:

"*Somehow similar patterns of biases also occur when looking at the standard deviation of the temperatures (Figure SM20)*"
- Lines 413-414 of the track-changes version and 399-400 of the revised one:
"*Biases in standard deviations for conditional precipitation values are also given for information (Figures SM29) and coincide with results for means.*"

With all the additional SM-figures, the revised supplementary material document now contains 37 figures. We believe that this is already a high number and thus we would prefer not to add other materials, especially with minor informative content.

Regarding why the Pearson correlation has been preferred over the Spearman one, the following text has been added to express our reasoning (lines 421-428 of the track-changes manuscript and lines 407-413 of the revised one):

*"Note that the Spearman rank correlation analysis provides similar conclusions, although they are perturbed by the rare rainfall occurrences, especially over North-Africa (not shown), which complicates the analysis of the boxplots. Indeed, daily precipitation data contain many zeros and therefore many tied first ranks. In a case with, for example, 100 values whose 80% are zeros, 80 ranks are ties and equal to to 1, while the first rank not equal to 1 is the $81^{st}$, creating a "jump" in the rank distribution. A relatively small error in the rainfall occurrence frequency can then lead to a high bias in the Spearman (rank)*
*correlation, while the Pearson correlation is less sensitive since it is based on "real" values and not ranks. Hence, the Pearson correlation has been preferred."*

comments on the text readability:

*The authors manage to describe quite clearly and elegantly their approach throughout the paper, with some minor readability*
*flaws: From line 142 the concept of "Block-A" and "Block-K" is explained. The explanation of the "Block-K" concept and why it is necessary could be a bit more elaborated. In line 189 and further down in the text the word "recopy" is used. What does it mean? How is it different from "copy"?*

Answer:

We would like to thank the reviewer for her nice compliments. We tried to improve the text regarding the concept of Block-A and Block-K in the revised manuscript. To do so, the text now includes (on lines 151-157 of the track-changes manuscript and lines 147-152 of the revised one):

*"Moreover, in order to avoid discontinuities in the reconstructed final sequence of ranks, the whole sequence of the best ana-logue is not fully kept, but only a sub-sequence corresponding to a given number of elements at the end of the complete*
*sequence. This kept sub-sequence is referred to "Block-K" (for "Block-kept") hereafter, and its length has also to be chosen, shorter or equal to Block-A. Searching for the best analogue with a length Block-A and then keeping only a length Block-K – shorter than Block-A – allows not only avoiding discontinuities in the (rank and correction) time series, but also giving flexibility to the proposed BC method to adapt to the temporal dynamics of the climate model to correct."*

Moreover, there is no difference between "copy" and "recopy". Hence, for clarification, we now use "copy" in the revised
manuscript.

technical corrections:

Comment:

*- line 16 gazes -> gas*

Answer:

Corrected.

Comment:

*- line 18 use of "climate change" and "climate changes" in the same sentence. Clarify difference between these terminologies or rewrite*

Answer:

The sentence has been modified, see lines 18-19:

*"... and are widely used by the scientific community investigating climate changes and their manifold impacts."*

Comment:

*- line 142 the sentence starting with "Moreover" is a bit convoluted*

Answer:

The sentence has been modified (on lines 151-157 of the track-changes manuscript and lines 147-152 of the revised one):

*"Moreover, in order to avoid discontinuities in the reconstructed final sequence of ranks, the whole sequence of the best ana-logue is not fully kept, but only a sub-sequence corresponding to a given number of elements at the end of the complete*

*sequence. This kept sub-sequence is referred to "Block-K" (for "Block-kept") hereafter, and its length has also to be chosen, shorter or equal to Block-A. Searching for the best analogue with a length Block-A and then keeping only a length Block-K – shorter than Block-A – allows not only avoiding discontinuities in the (rank and correction) time series, but also giving flexibility to the proposed BC method to adapt to the temporal dynamics of the climate model to correct."*

Comment:

*- line 217 you never use the term "copula effect" again consider adding links to the respective subfigures in section 4.1.2 for clarity and easy lookup*

Answer:

Indeed. As this term was never used again, the sentence *"In the following, we will refer to this mechanism as the "copula effect"."* has been deleted with no harm.

Comment:

*- line 232 hose -> those*

Answer:

Corrected.

Comment:

*- line 268 & section 4.2 do reference temporality and model chronology refer to the same thing? If so, consider only using one of the terms*

Answer:

No, the "reference temporality" refers to the temporal properties (e.g., auto-correlations) of the reference dataset. This is now
clarified in the track-changes manuscript (lines 296-297) and the revised one (lines 290-291):

"*... can present some underestimation of the temporal properties of the reference dataset, both for temperature and precipitation.*"

Comment:

*- line 437 particularly clear and concise summary*

Answer:

Thank you! We appreciate.

**References**

Dai, A.: Precipitation Characteristics in Eighteen Coupled Climate Models, Journal of Climate, 19, 4605–4630, https://doi.org/10.1175/JCLI3884.1, https://doi.org/10.1175/JCLI3884.1, 2006.

Kjellström, E., Boberg, F., Castro, M., Christensen, H., Nikulin, G., and Sánchez, E.: Daily and monthly temperature and precipitation statistics as performance indicators for regional climate models , Climate Research, 44, 135–150, https://www.int-res.com/abstracts/cr/v44/n2-3/p135-150/, 2010.

Teutschbein, C. and Seibert, J.: Bias correction of regional climate model simulations for hydrological climate-change impact studies: Review and evaluation of different methods, Journal of Hydrology, 456-457, 12 – 29, https://doi.org/https://doi.org/10.1016/j.jhydrol.2012.05.052, http://www.sciencedirect.com/science/article/pii/S0022169412004556, 2012.

---

## Author Comment (AC3) · 14 Sep 2020

**Responses to reviewer 2 (Sylvie Parey) about "R$^2$D$^2$: Accounting for temporal dependences in multivariate bias correction via analogue ranks resampling"**

Mathieu Vrac and Soulivanh Thao

Laboratoire des Sciences du Climat et de l'Environnement (LSCE-IPSL), CEA/CNRS/UVSQ, Université Paris-Saclay
Centre d'Etudes de Saclay, Orme des Merisiers, 91191 Gif-sur-Yvette, France

General comment:

*The paper handles the difficult issue of climate model bias correction extensions to tackle the adjustment of temporal, spatial and inter-variable dependency biases. Based on a previously proposed technique by the authors, named R2D2, different variants are designed and tested with one climate model simulation and one reference dataset for temperature and precipitations.*

*The methodology is meaningfully exposed and the results are clearly commented. This constitutes an important and valuable contribution to this question of bias correction, which remains a key issue in climate impact studies. Especially, the temporal evolution of the variables is often an important feature for climate impact models, which even require a finer timestep than daily (up to hourly).*

Answer:

We thank Dr. Sylvie Parey for her detailed review, compliments and constructive remarks. We tried to incorporate them in the revised manuscript.

Comment:

*My main comments are the following:*

*- When using CDFt for the univariate bias correction of rainfall, how are no-rain days handled? This may explain the degradation in the temporal autocorrelation after 1dBC, which is not seen for temperature. The adjustment of the number of rainy days, besides that of the rainfall amount, is one of the main problems in impact studies.*

Answer:

That is a relevant question since, indeed, a specific method (SSR, Vrac et al., 2016) is used to correct precipitation data and handle no-rain days. This method is now briefly detailed (lines 106-110 of the track-changes manuscript and 104-108 of the revised one) as follows:

"*A specific version of CDF-t is used to correct precipitation data. This version relies on a "Stochastic Singularity Removal"*

*(SSR, Vrac et al., 2016) approach to manage dry time steps: first, 0's (from both references and simulations) are randomly*

*transformed to positive but very small values ($< 10^{-6}$); then CDF-t is applied onto the whole set of data (i.e., transformed data and initially positive values altogether); and the correction results are thresholded such that values $< 10^{-6}$ are put to 0."*

Moreover, this remark leads to the question of the analyses of precipitation in separating rainfall occurrences and conditional precipitation intensities given rain occurrence, at least for marginal properties. Indeed, previously (i.e., in the initially submitted article), the univariate assessments of precipitation were done without separating occurrence and intensity. Hence, the analyses for the marginal of precipitation have been redone but, this time, separating precipitation occurrences and conditional precipitation intensities given rainfall occurrence. To do so, previous figure 7 (corresponding to boxplots of relative differences of mean precipitation with respect to the mean reference precipitation) is now replaced by:

– a figure showing boxplots of the relative differences of rainfall occurrence probability with respect to that of the reference;

– and, in the same figure, the boxplots of the relative differences of the mean conditional rain intensity (given rainfall occurrence) with respect to that of the reference dataset.

Note that, for each new boxplot figure, the associated maps are also given for each season as supplementary materials. In addition, boxplots of relative differences of the standard deviations computed from conditional precipitation intensities are also given in new figures of the supplementary materials.

To account for these changes, the text now reads (on lines 400-414 of the track-changes manuscript and lines 387-400 of the revised one):

*"For precipitation, univariate biases are investigated in separating occurrences of rainfall and conditional intensities given rainfall occurrences. Hence, Figure 7 displays, for the four seasons, boxplots of the relative differences of the probabilities of rainfall occurrence with respect to that the reference data (7(a-d)), as well as the boxplots of relative differences of the mean conditional intensities given rainfall occurrences, wrt that of the reference data (7(e-h)). The associated maps are given as supplementary materials (Figures SM21-24 for occurrence probabilities and SM25-28 for mean conditional intensities). Rainfall occurrences are defined as precipitation values $> 0.1mm/day$ to get rid of the drizzle effect present in many climate model simulations (e.g., Dai, 2006; Kjellström et al., 2010; Teutschbein and Seibert, 2012). Generally speaking, the effects of $R^2 D^2$ on the occurrence and conditional mean precipitation biases are similar to those observed on the mean temperature: (i) the R.1.1.0 configuration provides similar performances as BC1D; (ii) with or without time lags, and with or without adding precipitation in the conditioning, increasing the number of conditioning sites may lead to relatively higher biases, both for occurrence probability and intensity. However, including precipitation itself in the conditioning does not amplify the precipitation biases and can even reduce them depending on the season.*

*Biases in standard deviations for conditional precipitation values are also given for information (Figures SM29) and coincide with results for means."*

Comment:

*- Section 4.2 discusses the rank association between the corrected and the raw model simulation outputs. Indeed, there is no reason why the model should reproduce the observed chronology, since it represents another sequence of variability. However, some level of correction of the model chronology may be needed (in association with the variables interdependencies for example, as treated here). The problem then is the lack of an adequate reference...*

Answer:

The observed chronology (or at least the chronology of the reference dataset) is the only leverage point that we have to perform a correction of the model chronology. It is true that, as the model might behave quite differently from the reference in terms of temporality (i.e., has biases in its temporal properties), the rank association approach may not always be sufficient to fully correct the chronology-related aspects of the model, e.g., if the rank sequences of the raw model simulations are too unrealistic (i.e., too different from reference rank sequences). However, as the resampling is performed on observed rank sequences (supposed adequate), by construction, the $R^2D^2$ corrections including lags in the analogues will improve the temporal properties. This emphasises, nevertheless, the fact that correcting climate models that are too much biased may not be appropriate and provide non-negligible residual biases, here in the temporal properties.

Comment:

*- In section 4.3.2, it is noted that "the empirical copula between temperature and precipitation is not exactly the same during the two time periods used alternatively for calibration and validation". This may be worth exploring further: how stable is the association with time? How long has the considered period to be in order to faithfully estimate the association? How can climate change alter it? This problem is properly mentioned in the discussion, but could be raised here.*

Answer:

Those questions are indeed important. However, they are not specific to our suggested method and concern many methods and studies relying on dependencies of climate variables in climate change contexts. Actually, there is no clear attempt to tackle those questions in the scientific literature so far. Hence, we decided not to raise those problems in section 4.3.2, but to empha- sise them as discussions in section 5.2. We believe indeed that such fundamental questions deserve to be investigated on their own, and are thus left for future works.

The following paragraph – taking up the relevant questions of the reviewer – has then be added lines 509-514 of the track-changes manuscript and lines 494-499 of the revised one):

"*Hence, the potential non-stationarity of the dependencies between climate variables (i.e., rank association) may be worth*

*exploring further. This could be done via several questions, e.g.: how stable is the rank association with time? How long has the considered period to be in order to faithfully estimate the association? How can climate change alter it? Those questions, however, are not specific to our suggested method. They concern many methods and studies relying on dependencies of climate variables in climate change contexts. Such fundamental questions deserve to be investigated on their own, and are thus left for*

*future works."*

Comment:

*- In the conclusion, the improvements in the temporal evolution brought by the variants of R2D2 proposed here are emphasized, but still, they do not reach the rather good level obtained with 1dBC. Hence, in many impact studies, a correct chronology of the variables is of major interest, especially if long duration occurrences of extreme conditions are of concern. This could be*
*stated more clearly.*

Answer:

Indeed, for temporal properties of temperature, the CDF-t 1d-BC does a good job, due to the correct auto-correlations from the raw temperature simulations. However, this is not always that clear for precipitation, depending on the season. To state this
point more clearly, an additional text is now included lines 483-458 of the track-changes manuscript and lines 468-470 of the revised one):

"*For temporal properties, although the $R^2D^2$ variants strongly improve the initial $R^2D^2$ approach, they do not reach the rather good level obtained for temperature with the tested 1d-BC (due to the correct auto-correlations from the raw temperature simulations), while the results are more debatable for precipitation. In general, the main conclusions were that...*"

Minor comments:

Comment:

*- p2 l36: "whose the target is the whole univariate distribution": I would write "whose target"; the same applies to p7 l184, p9 l263*

Answer:

All are corrected.

Comment:
*- p4 l101: ""successive conditional": too many quotes*

Answer:

Corrected.

Comment:

*- p8 l232: "as hose from the IPSL dataset" as those*

Answer:

Corrected.

- *10 l302: "Finally, When adding time lags in the conditioning dimensions": no capital letter needed for "when"*

Answer:

Corrected.

Comment:

- *p10 l302-303: "both for temperature and precipitation, (R.1.1.0,R.1.5.0, R.1.100.0, R.2.1.0, R.2.5.0, R.2.100.0): isn't it rather R.1.1.1, R.1.5.1, R.1.100.1, etc... ? The same stands for the following sentence: "This is especially true, for R.2.5.0," R.2.5.1?*

Answer:

Absolutely! Thank you. This is now corrected.

Comment:

- *p13 l378: "We then compare these correlation values with them from the references." With those from the reference?*

Answer:

Corrected.

Comment:

*Supplementary Material:*

*Blank lines appear in some maps, what does it mean?*

Answer:

We do not see any blank lines in the supplementary materials figures. This may come from the pdf reader of the reviewer.

When zooming on the figures, some blank lines can sometimes also appear or disappear, depending on the pdf reader.

Comment:

*The scales in figures 17, 18 and 19 differ, which make it difficult to compare. This is the same for the precipitation figures. It may however be difficult to use the same scale for all plots, but when possible, it's easier for the reader.*

Answer:

We are aware that similar figures for different seasons may not have the same scale. This is a choice that we made to focus on the comparison between the raw, 1d-BC and the $R^2D^2$ variants. Indeed, since results for some seasons might vary very little compared to other ones, imposing a common scale to all seasons considerably resulted in very uniform maps and thus reduced the possibility to compare the methods. As our goal is not to compare between seasons but rather to compare the methods conditionally on specific seasons, scales were adapted to each season. However, we believe that some comparisons between seasons can be made anyway by the interested reader.

**References**

Dai, A.: Precipitation Characteristics in Eighteen Coupled Climate Models, Journal of Climate, 19, 4605–4630, https://doi.org/10.1175/JCLI3884.1, https://doi.org/10.1175/JCLI3884.1, 2006.

Kjellström, E., Boberg, F., Castro, M., Christensen, H., Nikulin, G., and Sánchez, E.: Daily and monthly temperature and precipitation statistics as performance indicators for regional climate models , Climate Research, 44, 135–150, https://www.int-res.com/abstracts/cr/v44/n2-3/p135-150/, 2010.

Teutschbein, C. and Seibert, J.: Bias correction of regional climate model simulations for hydrological climate-change impact studies: Review and evaluation of different methods, Journal of Hydrology, 456-457, 12 – 29, https://doi.org/https://doi.org/10.1016/j.jhydrol.2012.05.052, http://www.sciencedirect.com/science/article/pii/S0022169412004556, 2012.

Vrac, M., Noël, T., and Vautard, R.: Bias correction of precipitation through Singularity Stochastic Removal: Because occurrencesmatter, Journal of Geophysical Research: Atmospheres, 121, https://doi.org/10.1002/2015JD024511, 2016.